# Filling the Gaps: LLMs for Causal Hypothesis Generation

## Abstract

Scientific discovery catalyzes human intellectual advances, driven by the cycle of hypothesis generation, experimental design, data evaluation, and iterative assumption refinement. This process, while crucial, is expensive and heavily dependent on the domain knowledge of scientists to generate hypotheses and navigate the scientific cycle. Central to this is causality, the ability to establish the relationship between the cause and the effect. Motivated by the scientific discovery process, in this work, we formulate a novel task where the input is a partial causal graph with missing variables, and the output is a hypothesis about the missing variables to complete the partial graph. We design a benchmark with varying difficulty levels and knowledge assumptions about the causal graph. With the growing interest in using Large Language Models (LLMs) to assist in scientific discovery, we benchmark open-source and closed models on our testbed. We show the strong ability of LLMs to hypothesize the mediation variables between a cause and its effect. In contrast, they underperform in hypothesizing the cause and effect variables themselves. We also observe surprising results where some of the open-source models outperform the closed GPT-4 model.

## 1 Introduction

Scientific discovery has been key to humankind's advances. It is a dynamic process revolving around inquiry and constant refinements driven by new observations. Scientists adhere to a structured process that involves formulating a hypothesis and then collecting pertinent data (Wang et al., 2023a). They then draw inferences from experiments and the collected data, modify the hypothesis, formulate sub-questions, and repeat the process until the research question is answered (Kıcıman et al., 2023).

Causality empowers scientists to assess the hypotheses and interpret the collected data beyond mere correlations and associations. Tools such as Randomised Control Trials (RCTs) (Kendall, 2003) allow for establishing causal relationships between variables. Naturally, the process of causal discovery heavily relies on human experts to guide the hypothesis formation and experimental design (Kıcıman et al., 2023). Expert domain knowledge is crucial to narrow the search space of hypotheses, especially when it is expensive to collect data or when systematic exploration is infeasible. However, a possible impediment is that domain knowledge can be difficult to formulate and collect (Kıcıman et al., 2023).

With the recent advancement of Large Language Models (LLMs) (Brown et al., 2020; OpenAI, 2023), there has been a growing interest in using them for scientific discovery (AI4Science and Quantum, 2023; Lu et al., 2024; Cory-Wright et al., 2024). Their potential is now studied in domains such as natural sciences (AI4Science and Quantum, 2023). Given the importance of causality in the scientific discovery process, we focus on how LLMs can

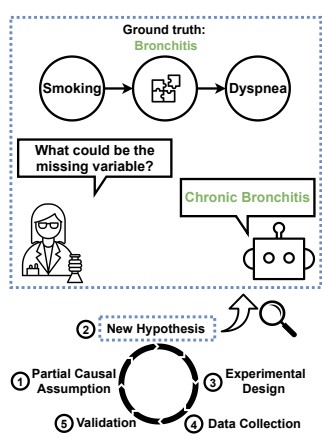

Figure 1: Scientific discovery iteratively generates hypotheses from assumptions using human expertise. We use LLMs as proxy experts to propose new hypotheses in causal DAGs.

assist with causal reasoning. LLMs have achieved state-of-the-art results for causal tasks such as determining pairwise causal relationships by considering variable names (Kıcıman et al., 2023), combined with causal discovery algorithms (Abdulaal et al., 2024; Ban et al., 2023a; Vashishtha et al., 2023) for refinement.

Causal discovery, however, comes *after* hypothesizing the variables of interest (which require domain knowledge), forming experiments, and potentially costly data collection. Our work, therefore, extends LLM applications to assist in steps essential *before* causal discovery, specifically identifying and hypothesizing *missing variables* in partially known causal graphs. This simulates the realistic scientific discovery process of incremental hypothesis formation and testing. By leveraging curated causal graphs, we evaluate the feasibility and reliability of LLMs in generating hypotheses under controlled yet realistic settings, ensuring reproducibility and providing a foundation for LLM-driven scientific discovery.

We break down causal hypothesis generation into smaller tasks, starting with baseline experiments, and progressing to realistic scenarios where only treatments and outcomes are known. We leverage LLMs' large-scale training to propose memorized or inferred variables based on their general and domain knowledge. This enables users to identify missing variables to guide data collection, followed by subsequent downstream causal tasks. Importantly, we avoid requiring LLMs to determine pairwise causal relations or perform numerical calculations, sidestepping their limitations in these areas (Zečević et al., 2023; Jin et al., 2023a). Existing work explores the inductive hypothesis generation capabilities of LLMs by using them as creative solution proposers with task-specific means of verifying said solutions (Romera-Paredes et al., 2023; Wang et al., 2023b; Qiu et al., 2024). In contrast, our work uniquely focuses on hypothesis generation within a causal paradigm.

**Contributions.** Our main contributions are: 1) We propose and formalize the novel task of LLM-assisted causal variable identification and hypothesizing. 2) We propose a benchmark for hypothesizing missing variables across diverse domains of existing causal graphs. 3) We design experimental tests with different difficulty levels and knowledge assumptions, such as open-world and closed-world settings, the number of missing variables, etc. 4) Our benchmark allows for allow for both grounded evaluations and a reproducible framework to benchmark LLMs' capabilities in hypothesis generation.

## 2 RELATED WORK

**LLMs and Causality.** Our work is based on the framework of causality as proposed by Pearl (2009). The intersection of language and causality is explored by Girju et al. (2002); Hassanzadeh et al. (2020); Tan et al. (2023); Dhawan et al. (2024) to extract causal relationships from a large corpus of text. With the advancements in LLMs and their ability to process large contexts, there has been an interest in using them for causal reasoning (Kıcıman et al., 2023). Some works have focused on commonsense causality (Frohberg and Binder, 2021; Singh et al., 2021) and temporal causal reasoning (Zhang et al., 2020; 2022). More recently Kıcıman et al. (2023); Long et al. (2023); Darvariu et al. (2024) introduced methods to discover causal structures by prompting LLMs with variable names. Ban et al. (2023b); Vashishtha et al. (2023); Ban et al. (2023a) extended this work by introducing ancestral constraints to refine the causal structures derived from LLMs. Abdulaal et al. (2024) combined data-based deep structural causal models, such as (Yu et al., 2019), with LLMs generated causal structure. Jin et al. (2023b) focused on causal inference using LLMs. While a tool on GitHub PyWhy-LLM used LLMs to propose confounders, our work formalizes such a task along with detailed insights. Recent work attempted to train transformers for improved causal inference and discovery (Vashishtha et al., 2024; Zhang et al., 2024). In contrast to previous work, we focus on the novel task of identifying and hypothesizing missing variables, a task that comes before data collection and evaluation, with LLMs as assistants. We test the hypothesizing abilities of generalist pre-trained LLMs as our task is primarily linked with pre-training knowledge.

**LLMs and hypothesis generation.** Existing works tested inductive hypothesis generation with LLMs in reasoning tasks or free-form scientific hypotheses from background knowledge provided in the context (Gendron et al., 2023; Qi et al., 2023; Xu et al., 2023a;b; Qiu et al., 2024; Lu et al., 2024). In contrast, we consider the structured task of causal hypothesis generation, where the ground-truth variables are known and can be used for evaluation. We also assume a pertinent human-in-the-loop assistive scientific discovery setup to counter LLMs' limitations and hallucinations.

## 3 PRELIMINARIES: CAUSAL GRAPH

A causal relationship can be modeled via a Directed Acyclic Graph (DAG). A causal DAG represents relationships between a set of $N$ variables defined by $\mathbf{V} = \{v_1, ..., v_N\}$. The variables are encoded in a graph $\mathcal{G} = (\mathbf{V}, \mathbf{E})$ where E is a set of directed edges between the nodes $\in$ V such that no cycle is formed. Mathematically it can be expressed as:

$$\mathcal{G} = (\mathbf{V}, \mathbf{E}), \ \mathbf{E} = \{e_{i,j} \mid v_i, v_j \in \mathbf{V}, i \neq j\} \text{ and } v_i \rightarrow v_j$$

Each edge $e_{i,j} \in \mathbf{E}$ denotes causal relationship between $v_i$ and $v_j$, $v_i \xrightarrow{e_{i,j}} v_j$, emphasizing the influence from $v_i$ to $v_j$. Beyond visualization, causal DAGs allow for the mathematical characterization of different node types for a causal model to understand the influences and dependencies.

We define $\mathbf{d}(v)$ as the degree of a node $v$, representing the total number of edges connected to $v$. $\mathbf{d}_{\text{in}}(v)$ is the in-degree, representing the number of incoming edges to $v$. $\mathbf{d}_{\text{out}}(v)$ is the out-degree, representing the number of outgoing edges from $v$.

**Sources** are variables $v_s$ with no incoming edges. Mathematically sources are $d_{\text{in}}(v_s) = 0$ where $d_{\text{in}}$ is the in-degree of the graph.
**Sinks** are variables $v_k$ with no outgoing edges. Sinks are $d_{\text{out}}(v_k) = 0$ where $d_{\text{out}}$ is the out-degree of the graph.
**Treatment** are variables $v_t$, characterized as nodes $d_{\text{in}}(v_t) = 0$ that are being intervened upon.
**Outcome** are variables $v_y$, characterized as the nodes $d_{\text{out}}(v_y) = 0$ that are observed for interventions from the treatments.
**Mediator** are variables $v_m$ that have both incoming and outgoing edges ($d_{\text{in}}(v_m) > 0$ and $d_{\text{out}}(v_m) > 0$), acting as intermediaries in the causal pathways between treatment and outcome.
**Confounder** are variables $v_k$ that influence both treatment and outcome, exhibiting edges directed towards the treatment and outcome nodes ($d_{\text{out}}(v_k) \geq 2$). Hence $v_k$ is a confounder if it is a parent of both $v_i$ and $v_j$.
**Collider** are variables $v_l$ that have two edges meeting, and have an in-degree greater than one $d_{\text{in}}(v_l) > 1$. Hence $v_k$ is a collider if it is a child of both $v_i$ and $v_j$.

**Average Treatment Effect.** Average Treatment Effect (ATE) quantifies the expected change in the outcome $v_y$ caused by the unit change of the treatment $v_t$. ATE is a part of the causal do-calculus introduced by Pearl (2009). We consider binary causal DAGs, i.e., each variable can either take 0 or 1 as values.

$$\text{ATE} = \mathbb{E}[v_y|\text{do}(v_t = 1)] - \mathbb{E}[v_y|\text{do}(v_t = 0)]$$

where the do($\cdot$) operator, represents an intervention. The $E[v_y|\text{do}(v_t = 1)]$ represents the expected value of the outcome variable $v_y$ when we intervene to set the treatment variable $v_t$ to 1 (i.e., apply the treatment), and $E[v_y|\text{do}(v_t = 0)]$ represents the expected value of $v_y$ when we set $v_t$ to 0 (i.e., do not apply the treatment).

**Mediation Analysis.** Mediation analysis is implemented to quantify the effect of a treatment on the outcome via a third variable, the mediator. The total mediation effect can be decomposed into the Natural Direct Effect (NDE) and the Natural Indirect Effect (NIE). The Natural Direct Effect (NDE) is the effect of the treatment on the outcome variable when not mediated by the mediator variable. The Natural Indirect Effect (NIE) is the effect of the treatment variable on the outcome variable when mediated by the mediator variable.

$$\text{NDE} = \mathbb{E}[v_{t=1}, v_{m=0} - v_{t=0}, v_{m=0}]$$

Here, NDE is calculated by comparing the expected outcome when the treatment variable is set to 1 and the mediator is fixed at the level it would take under the control treatment $v_t = 0$, with the expected outcome when both the treatment and the mediator are set to the control level.

$$\text{NIE} = \mathbb{E}[v_{t=0}, v_{m=1} - v_{t=0}, v_{m=0}]$$

Here, NIE is calculated by comparing the expected outcome when the treatment variable is set to 1 and the mediator is allowed to change as it would under the treatment, with the expected outcome when the treatment variable is set to 1 but the mediator is fixed at the control level.

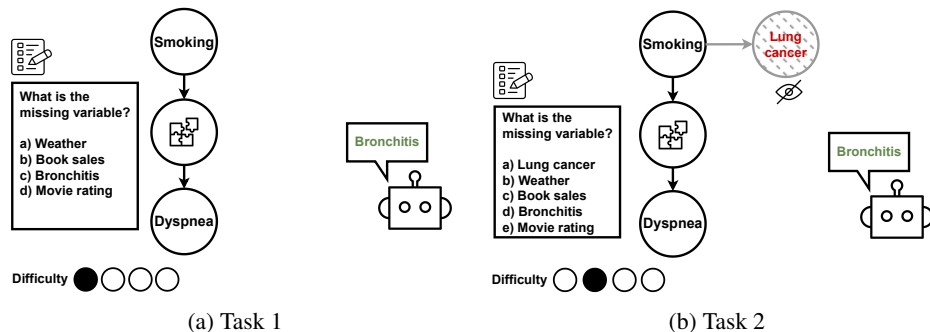

Figure 2: Leveraging LLM to indentify the missing variable for a causal DAG in the presence of out-of-context distractors (a), an in-context distractor along with out-of-context distractors (b).

# 4 LLMs for Identifying and Hypothesizing Causal Variables

In this work, we aim to leverage language models to identify and hypothesize variables in a causal DAG. Motivated by the process of hypothesizing a causal graph from a partially known structure Glymour et al. (2019), this paper proceeds under the assumption that some elements of the graph are already known. The aim is to find additional variables that can be incorporated into the existing causal structure to enhance the underlying causal mechanism.

We assume a partially known causal DAG, defined as $\mathcal{G}^* = (V^*, E)$, where $V^* \subseteq V$. The objective is to identify the set of missing variables $V^* = V \setminus V_{\text{missing}}$ thereby expanding $\mathcal{G}^*$ to $\mathcal{G}$. This implies that all causal relationships (edges) among variables in $V^*$ are known and correctly represented in $\mathcal{G}^*$; i.e., E is fully specified. Here, "missing" variables are not latent or hidden by measurement error but known unknowns within the causal graph reflective of LLMs perspective.

Our methodology is structured around progressively challenging scenarios, explores the ability of LLMs to identify and hypothesize causal variables. This starts from a restrictive and controlled exploration to an open-ended one. Initially, we restrict the exploration by providing the language models with a partially known causal DAG and a set of multiple choices for the missing variables. The complexity of the task is gradually increased by removing more than one node from the graph. Finally, we move to an open-ended scenario where the ground truth is not available to LLM. In this setting, LLM is required to hypothesize the missing variables of the causal DAG without any explicit hints. We evaluate the causal reasoning capability of LLMs through prompting. Given LLMs' limitation to textual input, we represent the graph $\mathcal{G}^*$ using a prompt template $P_{\text{LLM}}(\cdot)$ which enables LLMs to parse the causal relationships embedded within the DAG.

## 4.1 Task 1: Out-of-Context Controlled Variable Identification

This task (depicted in Figure 2a) evaluates LLMs' ability to identify missing variables in a causal graph from a list of multiple choices, thereby reconstructing the original graph. The partial DAG $\mathcal{G}^*$ is created by removing one variable from the original DAG $\mathcal{G}$. Let us denote the removed node as $v_x$. Along with the partial graphs, we operate in the multiple-choice question answering (MCQA) paradigm. The role of the LLM is to select a variable from the multiple choices, $\text{MCQ}_{v_x}$, that can be used to complete the graph. The multiple choices include the missing variable $v_x$ and out-of-context distractors. The out-of-context distractors are unrelated to the causal domain of the given DAG, chosen to minimize any contextual and overlap with the true missing variable. Let $v_x^*$ represent the variable selected by the LLM to complete $\mathcal{G}^*$.

$$v_x^* = P_{\text{LLM}}(\mathcal{G}^*, \text{MCQ}_{v_x}) \quad \forall v_x \in V$$

## 4.2 Task 2: In-Context Controlled Variable Identification

In practical applications, such as healthcare (Robins, 1986) and finance (Hughes et al., 2019), dealing with missing data and unobserved latent variables is a major challenge (Tian and Pearl, 2012; Bentler, 1980). Therefore, it is important to identify the missing variables and their underlying causal

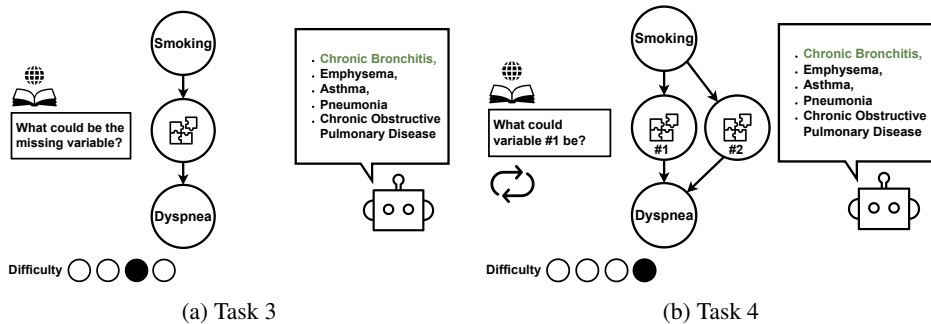

(a) Task 3           (b) Task 4

Figure 3: Leveraging LLM to hypothesize the missing variable in a causal DAG in an open-world setting for one variable (a), in an iterative fashion for multiple missing mediators (b).

mechanism. To simulate this, a more challenging task is introduced (see Figure 2b). Here, instead of removing one node from the ground truth DAG $\mathcal{G}$, two nodes, $v_{x_1}$ and $v_{x_2}$, are now removed to create the partial graph, $\mathcal{G}^*$.

$$\mathcal{G}^* = \mathcal{G} \setminus \{v_{x_1}, v_{x_2}\} \quad \text{for} \quad v_{x_1}, v_{x_2} \in \mathbf{V}$$

We use the MCQA paradigm to provide multiple choices that include the missing variables $v_{x_1}$ and $v_{x_2}$. The task for the LLM here is to select the correct variable $v_{x_1}$ only, given an in-context choice $v_{x_2}$ and out-of-context choices. The in-context variables are plausible variables within the same causal graph. We introduce the non-parental constrain for $v_{x_1}$ and $v_{x_2}$. This prevents the removal of both a parent node and its immediate child node in $\mathcal{G}^*$.

$$v_{x_1}^* = P_{\text{LLM}}(\mathcal{G}^*, \text{MCQ}_{v_{x_1}, v_{x_2}}) \ \ \forall v_{x_1}, v_{x_2} \in \mathbf{V} \text{ and } v_{x_1} \not\rightarrow v_{x_2}, \ v_{x_2} \not\rightarrow v_{x_1}$$

### 4.3 Task 3: Hypothesizing in Open World

So far, we have described the testbeds for variable identification in a partial DAG given the controlled world knowledge in the form of distractors. This assumption allows for the evaluation of the language model's ability to select the correct answer from a set of options. However, in the open-world setting, we increase the complexity to provide no choices, as shown in Figure 3a. Hence the task is to predict the missing node $v_x$ given the partial graph $\mathcal{G}^*$ to complete the ground truth graph $\mathcal{G}$. Here, the model returns a set of potential hypotheses, $\{v_{x,1}^*, ..., v_{x,k}^*\}$ where $k$ is the number of hypotheses.

$$\{v_{x,1}^*, v_{x,2}^*, ..., v_{x,k}^*\} = P_{\text{LLM}}(\mathcal{G}^*) \ \forall v_x \in \mathbf{V}$$

### 4.4 Task 4: Iteratively Hypothesizing in Open World

In addition to the search space relaxation, we further relax the number of missing variables. The partial DAG here, is obtained for one or more missing node variables. $\mathcal{G}^* = \mathcal{G} \setminus \{v_{x_1} ... v_{x_M}\}$. The fine-grained results from the open-world setting reveal that language models exhibit a particularly strong performance in identifying mediator variables. Thus, the LLM is used here to iteratively hypothesize mediator variables in a causal DAG given a treatment and an effect. The task (shown in Figure 3b) is set up as follows: given a partial graph $\mathcal{G}^*$, which includes observed treatment and outcome variables, we aim to hypothesize a set of mediators, denoted as $M = \{v_{m_1}, v_{m_2}, ..., v_{m_H}\}$, that mediates the treatment $v_t$ to the outcome $v_y$. Here, $H$ represents the number of direct, and indirect mediators. A pair of treatments and outcomes are considered iteratively across the causal DAG. In the first iteration, the LLM generates a hypothesis for the mediator $v_{m_1}$. The hypothesized mediator, $v_{m_1}$ is then added to the graph, updating $\mathcal{G}^* \rightarrow \mathcal{G}^* \cup \{v_{m_1}\}$. The partial graph that now also includes $v_{m_1}^*$ can be used to identify the second mediator $v_{m_2}^*$ and so on. Therefore, in each subsequent iteration $i$, the LLM is tasked to generate a hypothesis for the next missing mediator $v_{m_i}$ given the updated graph $\mathcal{G}^* \cup \{v_{m_1}^*, ..., v_{m_{i-1}^*}\}$.

$$v_{m_i}^* = P_{\text{LLM}}(\mathcal{G}^* \cup \{v_{m_1}^*, ..., v_{m_{i-1}}^*\}) \text{ for } i = 1, ..., H$$

The sequence of mediators $M = \{v_{m_1}, v_{m_2}, ..., v_{m_H}\}$ is chosen at random. To formally investigate how the order of hypothesized mediators influences LLM performance, we borrow concepts from the

mediation analysis literature, specifically the Natural Direct Effect (NDE) and the Natural Indirect Effect (NIE). The NDE measures the effect of the treatment on the outcome that is not mediated by a particular mediator, while the NIE measures the effect of the treatment that is mediated by the mediator. We introduce a metric called Mediation Influence Score (MIS) that quantifies the influence of each mediator between a treatment and an effect. MIS defined as the ratio of NIE to NDE, provides a scale-free measure of a mediator's relative influence, enabling prioritization. MIS is always positive, reflecting the absolute contribution of mediators.

$$\text{MIS}\,(v_{m_i}) = \left| \frac{\text{NIE}(v_{m_i})}{\text{NDE}(v_{m_i})} \right| \quad \text{for} \quad i = 1, ..., H.$$

This metric quantifies the relative importance of the indirect effect (through the mediator) compared to the direct impact. Mediators are then ranked and prioritized based on their MIS scores, with higher scores indicating a stronger mediation effect.

## 5 EVALUATION AND RESULTS

### 5.1 EXPERIMENTAL SETUP

We evaluate a variety of causal datasets spanning diverse domains. We use the semi-synthetic datasets from BNLearn repository - Cancer:$\mathcal{G}(5, 4)$ (Korb and Nicholson, 2010), Survey:$\mathcal{G}(6, 6)$ (Scutari and Denis, 2021), Asia:$\mathcal{G}(8, 8)$ (Lauritzen and Spiegelhalter, 1988), Child:$\mathcal{G}(20, 25)$ (Spiegelhalter, 1992), Insurance:$\mathcal{G}(27, 52)$ (Binder et al., 1997), and Alarm:$\mathcal{G}(37, 46)$ (Beinlich et al., 1989). We also evaluate our approach on a realistic Alzheimer's Disease dataset:$\mathcal{G}(9, 16)$ (Abdulaal et al., 2024), developed by five domain experts. We also test on a legal causal graph, Law:$\mathcal{G}(8, 20)$ (VanderWeele and Staudt, 2011). See Appendix A.1 for further details.

We evaluate our setups across different open-source and closed models. The models we use are GPT-3.5 (Brown et al., 2020), GPT-4 (OpenAI, 2023), LLama2-chat-7b (Touvron et al., 2023), Mistral-7B-Instruct-v0.2 (Jiang et al., 2023), Mixtral-7B-Instruct-v0.1 (Jiang et al., 2024), Zephyr-7b-Beta (Tunstall et al., 2023) and Neural-chat-7b-v3-1 (Intel, 2023).

Implementation details are mentioned in Appendix A. Prompt templates are illustrated in Appendix F.

### 5.2 TASK 1: OUT-OF-CONTEXT CONTROLLED VARIABLE IDENTIFICATION

Our first experiment is designed to assess the fundamental ability of language models to identify missing variables in a partial causal graph, serving as a baseline for understanding their performance in variable identification tasks. Here, the input to the LLM is the ground truth variable name in addition to out-of-context multiple choices for the missing variable $v_x$ and the partial DAG $\mathcal{G}^*$. We then calculate the models' accuracy in correctly predicting $v_x$.

$$\text{Accuracy} = \frac{1}{N} \sum_{i=1}^{N} \mathbb{1}(v_x^* = v_x^i)$$

**Results.** In Figure 4a, we report the accuracy of different LLMs in identifying the missing variable. GPT-4, followed by Mixtral, consistently performs well, achieving perfect accuracy on most of the datasets. GPT-3.5 also shows overall strong performance, apart from the Insurance and Alarm datasets. The other models, including Mistral-7b, Llama-7b, and Zephyr-7b, demonstrate varying degrees of success. Insurance is the most challenging dataset, which could potentially be due to the high number of edges present in the DAG. It is noteworthy that all models significantly outperform the random baseline, indicating that given out-of-context multiple choices along with the ground truth variable, the language model can pick out the missing causal variable to complete the partial graph $\mathcal{G}^*$. However, we may conjecture that the high performance could be attributed to the simplicity of the task. The models might be primarily inferring from the context of the dataset domain, rather than performing actual causal reasoning among multiple plausible choices. To further investigate this, we introduce an in-domain choice in the multiple choices in the next experiment. This can assess LLMs' ability to choose a causal variable for a partial DAG beyond the highly evident correlations.

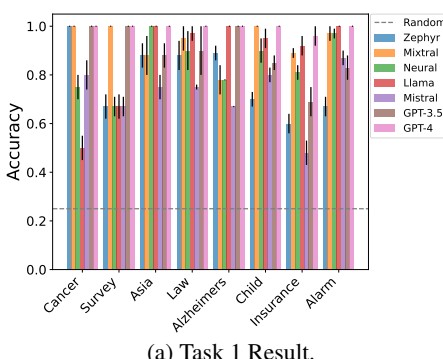 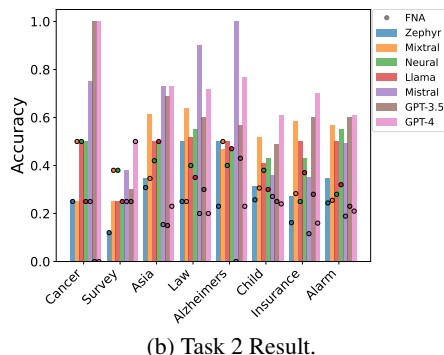

(a) Task 1 Result.          (b) Task 2 Result.

Figure 4: Accuracy of LLMs in identifying the missing causal variable from multiple choices with out-of-context distractors (a), and from both out-of-context and in-context distractors (b).

| | Cancer | | Survey | | Asia | | Law | | Alzheimers | | Child | | Insurance | | Alarm | | Avg | |
| | Sim | LLM-J | Sim | LLM-J | Sim | LLM-J | Sim | LLM-J | Sim | LLM-J | Sim | LLM-J | Sim | LLM-J | Sim | LLM-J | Sim | LLM-J |
|---|---|---|---|---|---|---|---|---|---|---|---|---|---|---|---|---|---|---|
| Zephyr | 0.36 | 0.61 | 0.34 | 0.60 | 0.45 | 0.66 | 0.41 | 0.70 | 0.35 | 0.75 | 0.51 | 0.70 | 0.45 | 0.44 | 0.46 | 0.69 | 0.42 | 0.63 |
| Mixtral | 0.41 | 0.66 | 0.39 | 0.66 | **0.66** | 0.75 | 0.38 | 0.69 | 0.31 | 0.77 | **0.53** | **0.77** | 0.46 | **0.56** | **0.50** | 0.72 | 0.46 | 0.70 |
| Neural | 0.38 | 0.77 | 0.43 | 0.55 | 0.53 | 0.55 | 0.47 | 0.72 | 0.44 | 0.71 | 0.48 | 0.70 | 0.47 | 0.43 | 0.47 | 0.67 | 0.45 | 0.63 |
| Llama | 0.40 | 0.48 | 0.40 | 0.54 | 0.53 | 0.58 | 0.67 | 0.65 | 0.45 | 0.61 | 0.48 | 0.63 | 0.42 | 0.34 | 0.46 | 0.65 | 0.45 | 0.55 |
| Mistral | 0.33 | 0.67 | 0.44 | 0.65 | 0.60 | 0.73 | 0.49 | 0.67 | 0.34 | 0.76 | 0.48 | 0.68 | 0.46 | 0.47 | 0.47 | 0.71 | 0.44 | 0.67 |
| GPT-3.5 | 0.48 | 0.74 | 0.42 | **0.79** | 0.47 | 0.61 | 0.52 | 0.73 | 0.39 | **1.00** | 0.36 | 0.60 | 0.47 | 0.52 | 0.48 | 0.73 | 0.44 | 0.71 |
| GPT-4 | **0.49** | **0.90** | **0.51** | 0.67 | **0.66** | **0.76** | 0.55 | **0.78** | **0.47** | 0.98 | 0.36 | 0.53 | **0.52** | **0.56** | 0.49 | **0.75** | **0.50** | **0.73** |

Table 1: Task 3 Results. Average semantic similarity and LLM-as-Judge metrics to evaluate LLMs in hypothesizing the missing variable in a causal DAG.

## 5.3 TASK 2: IN-CONTEXT CONTROLLED VARIABLE IDENTIFICATION

We introduce a more complex setting to further challenge the models' abilities in missing variable identification. In this setup, the partial graph has two missing nodes. Alongside out-of-context choices and the ground truth variable, the multiple-choice options also include the second missing node from the partial graph as an in-context distractor. This configuration requires the language model to reason about indirect causal relationships to identify the correct missing variable. To evaluate models' performance, we present two metrics: accuracy and false node accuracy. The false node accuracy, measures the confusion of LLMs in picking the in-context variable instead of the ground truth.

$$\text{False Node Accuracy (FNA)} \downarrow = \frac{1}{N} \sum_{i=1}^{N} \mathbb{1}(v_{x_1}^* = v_{x_2})$$

**Results.** In Figure 4b, we plot both Accuracy and False Node Accuracy across different datasets. Ideally, accuracy should be $1.0$, and the FNA should be $0.0$. Since there were 5 multiple choices, the random chance is $0.2$. We observe that most of the models for larger datasets achieve much higher accuracy than random chance. GPT-3.5 and GPT-4 consistently perform well across all datasets, with high accuracy and low FNA. This suggests that these models are capable of reasoning by identifying the missing nodes in the causal graph and are less likely to be confused by the in-context node variable. On the other hand, open-source models like Mistral, Zephyr, and Mixtral show varying performance across different datasets. For instance, Mistral performs well on the easy Cancer dataset but underperforms in the more complex Alarm dataset. In summary, we observe that most language models can identify causal variables in the presence of multiple missing nodes and an in-context distractor. These results indicate that while most language models can handle missing variable identification in the presence of multiple missing nodes and in-context distractors, the robustness of their reasoning abilities varies significantly with dataset complexity and model architecture.

## 5.4 TASK 3: HYPOTHESIZING IN OPEN WORLD

We recall that the goal is for the language models to be able to complete a causal graph given a partial graph. In realistic scenarios, where scientists provide incomplete graphs without pre-defined answers,

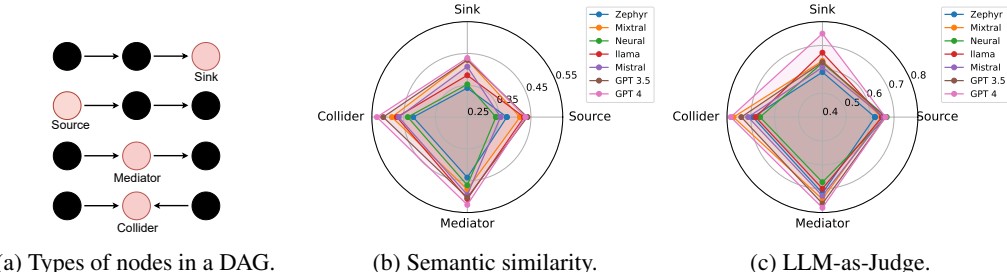

(a) Types of nodes in a DAG.    (b) Semantic similarity.    (c) LLM-as-Judge.

Figure 5: Task 3 Results. Visualizing each model's performances, averaged across the different datasets, for Sink, Source, Mediator, and Collider nodes.

there is often no single 'ground truth' for what the missing variable should be. The correct hypothesis may vary based on domain expertise or available data, making this task fundamentally open-ended. Hence in this test-bed, we aim to leverage LLMs to hypothesize the causal variables. The language model is prompted for $k = 5$ suggestions for the missing node $v_x$.

We then compare the suggestions against the ground truth, acknowledging that in realistic scenarios, there is often no single 'ground truth' for the missing variables. This complexity necessitates careful evaluation, as we suspect that traditional metrics may not fully capture the performance of models, particularly when suggestions must be assessed within the broader context of the entire causal graph (see Appendix C.5 for more details). Hence, for a robust evaluation of this experiment, we use two metrics, semantic similarity, and LLM-as-Judge that incorporate contextual information.

**Semantic Similarity:** measures the cosine similarity between the embeddings (of another pretrained sentence embedding model) of each suggestion of the model's predictions, $v_{x_{1:5}}^*$ and the ground truth $v_x$. The distances of the most similar suggestion are averaged across all nodes $v_x \in \mathbf{V}$. For a detailed explanation of this process, please refer to Appendix A.4.

**LLM-Judge:** This metric evaluates the quality of the model's suggestions using a two-step process inspired by Zheng et al. (2023). In particular, LLM-as-Judge compares against ground truth variables to measure contextual semantic similarity beyond semi-exact matching like in semantic similarity metric. In the first step, the language model is prompted to determine which suggestion best fits the partial graph, given the ground truth and the suggestions, $v_{x_{1:5}}^*$. In the second step, the language model is again prompted to rate the selected suggestion on a scale of 1 to 10 in terms of similarity. This is repeated for all nodes, and the ratings are averaged to provide an overall quality measure. Implementation details can be found in Appendix A.5.

**Results.** We report models' performances using both semantic similarity and LLM-Judge metrics in Table 1. For brevity, we provided the variances in Appendix C.1. To develop an intuition of LLMs' performance, we provide a detailed analysis of each metric across different types of node variables (defined in Section 3). We specifically look at sources, sinks, colliders, and mediators for each of the partial causal graphs. The results, fine-grained by node type, are given in Figure 5 that shows each model's average performance across datasets. For a detailed performance per each dataset individually, see Figure 18.

GPT-4 and Mistral generally achieve higher semantic similarity and LLM-as-Judge scores across most datasets (Figure 18). GPT-3.5 also shows good average performance. We observe that semantic similarity is a stricter metric than LLM-as-judge since it cannot encode contextual information about the causal DAG (see example in Table 7). Despite different scales, semantic similarity and LLM-as-judge metrics both seem to be fairly correlated. In Figure 5, we observe that models display stronger performance for colliders and mediators on average. This suggests that these models are relatively proficient at reasoning about common causes and indirect causal relationships. Sinks are typically the nodes that represent the outcomes or effects of interventions (treatments) applied to other nodes, and the lower performance on these nodes indicates that the models find it challenging to reason about the potential outcomes of the causal graphs. Source nodes represent the causes in a causal graph, and lower performance on these nodes might indicate difficulties in reasoning about potential treatments from the partial graph.

In Figure 16a, we observe that the model performance increases with $k$, i.e., with a higher number of suggestions. From Figure 16b, it is also evident that the performance is proportional to the number of total edges, $d_{in} + d_{out}$ (more context about the node). In summary, LLMs show impressive

performance across some of the nodes and can be particularly useful to hypothesize mediators and colliders in a partial causal DAG. It is, hence, potentially beneficial to use LLMs in the real world because, in practice, treatment and outcomes are usually known.

### 5.4.1 Hypothesizing Confounder

In causal inference, backdoor paths are alternative causal pathways that confound the estimation of causal effects. They introduce bias when estimating causal effects if not appropriately addressed. Hence hypothesizing and controlling for confounders is an important task in causal inference (Pourhoseingholi et al., 2012). We extract confounder subgraphs from (Sachs et al., 2005), Alarm, and Insurance graphs. From Table 2, with detailed results in Appendix B, we observe that while some confounders were easily hypothesized by LLMs, achieving perfect accuracy, the genomic domain of the SACHS posed challenges for models with potentially less domain-specific knowledge. Similar to the mediator analysis, a large model: GPT-4, does not always perform best across all datasets. This highlights the need for a diverse set of benchmarks, like ours, to fully assess models' performance. Considering the importance of backdoor paths, we have also benchmarked LLM performance for confounders in addition to colliders. LLM typically performs well when hypothesizing a collider, however, the results for confounders are varied.

|         | Sachs | Alarm | Ins |
|---------|-------|-------|-----|
| Zephyr  | 0.10 ±0.01 | 0.45 ±0.05 | 0.53 ±0.06 |
| Mixtral | **0.95** ±0.10 | **0.85** ±0.09 | 0.63 ±0.07 |
| Neural  | 0.30 ±0.03 | 0.45 ±0.05 | 0.61 ±0.06 |
| LLama   | 0.20 ±0.02 | 0.47 ±0.05 | 0.63 ±0.06 |
| Mistral | 0.20 ±0.02 | **0.85** ±0.09 | 0.61 ±0.06 |
| GPT-3.5 | 0.40 ±0.04 | 0.49 ±0.05 | 0.67 ±0.07 |
| GPT-4   | **0.95** ±0.10 | 0.73 ±0.07 | **0.78** ±0.08 |

Table 2: Hypothesizing Confounders.

### 5.5 Task 4: Iteratively Hypothesizing in Open World

In the previous open-world experiment, we observed that LLMs excel at identifying mediators when the treatments and outcomes are given. This observation could be particularly relevant in medical settings, where understanding the mediators can provide insights into causal mechanisms through which a treatment affects a patient's outcome.

For hypothesizing mediators, we adopted an iterative approach rather than a global (all-at-once) strategy. This interactive process allows the language model to progressively refine its predictions, reducing the search space for subsequent variables. As observed in our empirical results (see Appendix C.7), LLMs underperform when tasked with making multiple simultaneous predictions across different mediators. The iterative approach aligns more closely with human reasoning, as evidenced by Chain-of-Thought (CoT) (Wei et al., 2022) strategies, where sequential decision-making enhances accuracy.

For **unordered mediator evaluation**, the model is prompted iteratively with mediators presented in random order, and the final semantic similarity is averaged across all predictions. In contrast, **ordered mediator evaluation** ranks the mediators using the Mediation Influence Score (MIS), prompting the model in both ascending and descending orders of significance. We introduce the metric $\Delta$, presenting the difference in performance when mediators are iteratively presented to the LLM in ascending and descending orders of significance defined by the MIS. Given that some datasets only contain a single mediator, we selected the Asia, Child, Insurance, and Alarm datasets, as they offer a wider range of mediators, ranging from *1 to 10* for the Alarm dataset.

**Results.** The results of this experiment are in Table 3. Results with variances are provided in Appendix C.1. In a highly complex environment with more than one node missing and with open-world search space, we observe that LLMs can still maintain their performance. Unlike the overall consistent performance of GPT-4 across all of the datasets from the open-world setting, the model showed superior performance in Insurance and Alarm datasets only. As the complexity of the dataset increases, we observe larger differences in hypothesizing the mediators according to the MIS order. Positive $\Delta$ values suggest that prompting the LLM based on the MIS metric leads to higher semantic similarity between the mediator hypotheses and the ground truth variables. In summary, we observe that LLMs can be highly effective in iteratively hypothesizing multiple mediators in a DAG, and if present, some domain knowledge about the significance of the mediator can boost the performance.

|  | Asia | | Child | | Insurance | | Alarm | |
|---|---|---|---|---|---|---|---|---|
|  | Sim | $\Delta$ | Sim | $\Delta$ | Sim | $\Delta$ | Sim | $\Delta$ |
| Zephyr | 0.61 | $-0.02$ | **0.54** | 0.17 | 0.47 | 0.19 | 0.51 | 0.20 |
| Mixtral | **0.87** | 0.01 | 0.50 | 0.18 | 0.48 | 0.15 | 0.52 | 0.13 |
| Neural | 0.65 | 0.04 | 0.48 | 0.21 | 0.42 | 0.16 | 0.46 | 0.12 |
| Llama | 0.80 | 0.07 | 0.49 | $-0.05$ | 0.44 | 0.21 | 0.51 | 0.07 |
| Mistral | 0.33 | 0.02 | 0.50 | 0.12 | 0.48 | 0.13 | 0.47 | 0.11 |
| GPT-3.5 | 0.48 | 0.01 | 0.36 | 0.25 | 0.48 | 0.17 | 0.51 | 0.02 |
| GPT-4 | 0.49 | 0.04 | 0.39 | 0.16 | **0.52** | 0.14 | **0.60** | $-0.07$ |

Table 3: Sim: semantic similarity for iteratively hypothesizing the mediator nodes (Task 4) when prompted with random order. $\Delta$ measures the change in the prediction of each model when repeating the experiment with ordering according to the MIS metric instead of randomly.

## 5.6 DISCUSSION AND LIMITATIONS

The results show that LLMs effectively hypothesize missing variables, particularly mediators, though performance varies with task complexity. Simple tasks, like identifying missing variables from controlled options, had high success rates. Without unified metrics, we focused on relative rankings across models (Appendix C.2) and observed that no model, including GPT-4, consistently outperformed the others. We hypothesize that the differences in performance across domains may stem from potential biases within the LLMs. These biases may stem from the models' training data and, therefore, its parametric memory, leading to disparities in how effectively the models handle different tasks introduced in the benchmark. For instance, the models' ability to hypothesize confounders varied significantly across datasets. In some cases, such as the Sachs dataset (see Appendix B), domain-specific knowledge gaps may have led to lower accuracy.

While this paper aimed to evaluate the ability of current LLMs to identify and hypothesize variables in a partial causal graph, we attempted to improve the performance by fine-tuning the model and few-shot prompting. However, given the limited size of the DAGs used, the resulting datasets were small, leading to mixed results (see Appendix D.1). We suspect that while fine-tuning may help the model to specialize, it can also reduce its ability to leverage the general parametric knowledge (Yang et al., 2024). Future approaches can look at domain-specific fine-tuning.

Given the non-disclosed datasets of models, it is difficult to confirm with absolute certainty that the datasets are not ingested by models during training. However, one of the datasets we used was released recently (Abdulaal et al., 2024) after the announced cut-off date of models. Additionally, our task itself is novel, including the way we verbalize the graphs and prompt the models. Additionally, in Table 3 we further demonstrate that LLM-generated suggestions are non-verbatim, indicating they generate novel hypotheses rather than retrieving memorized patterns. Finally, we did not observe any direct reconstruction of graphs that would suggest memorization.

Our setup assumes known edges among missing variables to enable controlled evaluation, which future work can extend. We envision this as a human-LLM collaboration under expert supervision, as LLMs cannot automatically identify the most plausible answer or express confidence in their responses (Zhou et al., 2024). Future work could explore better filtering mechanisms and improve performance on source and sink nodes.

## 6 CONCLUSION

Most causality literature assumes that the necessary data has been collected, focusing on establishing causal relationships between variables. However, generating hypotheses about which variables to observe is typically done by human experts. LLMs, trained on large-scale datasets, can act as expert proxies for this task. We introduce the novel task of using LLMs to hypothesize missing variables in causal graphs, formalizing it with benchmarks that vary in difficulty and knowledge of the ground truth graph. We evaluate models on identifying missing variables from in-context and out-of-context distractors and hypothesizing variables in an open-world setting. We also explore an iterative approach for populating graphs with up to 10 missing mediator nodes. Our results show that LLMs are particularly effective at hypothesizing mediators, which are often less known than treatments and outcomes. To support further research, we will release our benchmark and codebase.

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

# A  IMPLEMENTATION

## A.1  DATASETS

We use 7 real-world based datasets. These datasets span different domain knowledge topics. These datasets have ground truth graphs along with their observational data. The simplest dataset used is the cancer dataset with 4 edges and 5 node variables. In addition to the semi-synthetic datasets from the BNLearn library, we also evaluate our approach on a realistic Alzheimer's Disease dataset Abdulaal et al. (2024), which was developed by five domain experts. Given that each expert created a different causal graph, the final causal DAG comprises only those edges that were agreed upon by consensus.

| Dataset | V | E | Description |
|---|---|---|---|
| Cancer | 5 | 4 | Factors around lung cancer |
| Survey | 6 | 6 | Factors for choosing transportation |
| Asia | 8 | 8 | Factors affecting dysponea |
| Law | 8 | 20 | factors around legal system |
| Alzheimer | 9 | 16 | Factors around Alzheimer's Disease |
| Child | 20 | 25 | Lung related illness for a child |
| Insurance | 27 | 52 | Factors affecting car accident insurance |
| Alarm | 37 | 46 | Patient monitoring system |

Table 4: Dataset description.

## A.2  REPRODUCIBILITY

For reproducibility, we used temperature 0 and top-p value as 1 across all of the models. We also mentioned the snapshot of the model used. We have also included the prompts and examples below. Our code can be anonymously found here - `https://anonymous.4open.science/r/causal-llm-env-6C8E/README.md`. The datasets are under CC BY-SA 3.0 which allows us to freely modify the datasets for benchmarking. Our benchmark will be released under the CC BY-SA License.

GPT-3.5 GPT-4 were accessed via API. Rest of the models were run on 1 A100 GPU. Since we used off-the-shelf LLM, there was no training to be performed. Since many of the models were run by API, it is difficult to calculate the entire compute, however, all of the experiments for each model took $\approx 6$ hours.

## A.3  CONTROLLED VARIABLE IDENTIFICATION

For variable identification, we generate multiple choices that remain consistent across all missing nodes and all of the datasets. The words were randomly chosen to be far enough from the nodes. The options chosen were weather, book sales, and movie ratings. We wanted to make sure that the options were not from one specific domain such that the LLM could do the process of elimination.

## A.4  SEMANTIC SIMILARITY

Given the task of hypothesizing missing nodes in a partial graph $\mathcal{G}^*$ in the absence of multiple-choices, we evaluate the semantic similarity between the model's predictions and the ground truth node variable. We leverage an open model namely 'all-mpnet-base-v2' to transform the textual representations of the model's predictions and the ground truth into high-dimensional vector space embeddings. Post transforming textual representations into embeddings and normalizing them, we calculate the cosine similarity. Scores closer to 1 indicate a high semantic similarity, suggesting the model's predictions align well with the ground truth. This metric gives a score of similarity without the contextual knowledge of the causal graph. We perform our experiments to consider every node of the ground truth as a missing node iteratively. For all the suggestions for a node variable, we calculate the semantic similarity. The average similarity reported is the highest semantic similarity for each of the variable suggestions.

---

**Algorithm 1** Evaluating Semantic Similarity for Hypothesized Missing Nodes

---

1: **Input:** Partial graph $\mathcal{G}^*$, Ground truth node variables $V_{\text{GT}}$, Language model $LM =$ 'all-mpnet-base-v2'
2: **Output:** Average highest semantic similarity score
3: **procedure** SEMANTICSIMILARITY($\mathcal{G}^*, V_{\text{GT}}, LM$)
4:    Initialize $similarityScores$ as an empty list
5:    **for** each node $v_{\text{GT}}$ in **v do**
6:       $predictions \leftarrow$ GeneratePredictions($\mathcal{G}^*, LM$)
7:       Initialize $nodeScores$ as an empty list
8:       **for** each prediction $p$ in $predictions$ **do**
9:          $embedding_{\text{GT}} \leftarrow$ Embed($v_{\text{GT}}, LM$)
10:          $embedding_p \leftarrow$ Embed($p, LM$)
11:          Normalize $embedding_{\text{GT}}$ and $embedding_p$
12:          $score \leftarrow$ CosineSimilarity($embedding_{\text{GT}}, embedding_p$)
13:          Append $score$ to $nodeScores$
14:       **end for**
15:       $maxScore \leftarrow$ Max($nodeScores$)
16:       Append $maxScore$ to $similarityScores$
17:    **end for**
18:    $averageScore \leftarrow$ Average($similarityScores$)
19:    **return** $averageScore$
20: **end procedure**

---

| **Ground Truth**: | Smoking status | | | | |
| *LLM Suggestions*: | Smoking | Alcohol Consumption | Exposure to Radiation | Poor Diet | Genetic Predisposition |
| Semantic similarity : | 0.72 | 0.38 | 0.22 | 0.22 | 0.17 |
| **Ground Truth**: | Employee or self-employed | | | | |
| *LLM Suggestions*: | Income Level | Job Location | Environmental Awareness | Lifestyle Preferences | Health Consciousness |
| Semantic similarity : | 0.30 | 0.25 | 0.17 | 0.15 | 0.10 |
| **Ground Truth**: | Dyspnea laboured breathing | | | | |
| *LLM Suggestions*: | Shortness of breath | Chest Pain | Coughing | Fatigue | Weight Loss |
| Semantic similarity : | 0.57 | 0.41 | 0.36 | 0.29 | 0.11 |
| **Ground Truth**: | Montreal Cognitive Assessment score | | | | |
| *LLM Suggestions*: | Cognitive Function | Neurological Function | Mental Health Status | Risk of Alzheimer's Disease | Memory Performance |
| Semantic similarity : | 0.60 | 0.47 | 0.38 | 0.36 | 0.16 |
| **Ground Truth**: | Grunting in infants | | | | |
| *LLM Suggestions*: | Respiratory distress | Asthma | Pneumonia | Pulmonary infection | Bronchopulmonary dysplasia |
| Semantic similarity : | 0.22 | 0.18 | 0.17 | 0.11 | 0.01 |
| **Ground Truth**: | Driving history | | | | |
| *LLM Suggestions*: | Previous accidents | Distance driven daily | Type of car insurance | Frequency of car maintenance | Location of parking |
| Semantic similarity : | 0.55 | 0.42 | 0.27 | 0.26 | 0.18 |
| **Ground Truth**: | Heart rate blood pressure | | | | |
| *LLM Suggestions*: | Pulse Rate | Blood Pressure | Respiratory Rate | EKG Reading | Blood Oxygen Level |
| Semantic similarity : | 0.78 | 0.78 | 0.57 | 0.49 | 0.42 |

Table 5: Examples of model suggestions from and the corresponding semantic similarity score for a missing node variable from each of the datasets.

## A.5 LLM-AS-JUDGE

To capture the domain knowledge of the expert that selects the most relevant causal variable, we use LLM-as-Judge as a proxy expert. This also allows for evaluation based on contextual DAG knowledge as well. Given the impressive results of GPT-4 in (Zheng et al., 2023), we use GPT-4 as a judge for all of the experiments.

---

**Algorithm 2** Evaluating Model Suggestions with LLM as Judge

1: **Input:** Partial graph $\mathcal{G}^*$, Ground truth node variables $V_{\text{GT}}$, Predictions $P$, Language model LLM = GPT-4
2: **Output:** Average quality rating of model's suggestions
3: **procedure** LLMASJUDGE($\mathcal{G}^*$, $V_{\text{GT}}$, $P$, LLM)
4:     Initialize $qualityRatings$ as an empty list
5:     **for** each node $v_{\text{GT}}$ in **V do**
6:         $suggestions \leftarrow$ GenerateSuggestions($\mathcal{G}^*$, $P$, LLM)
7:         $bestSuggestion \leftarrow$ SelectBestSuggestion($suggestions, v_{\text{GT}}$, LLM)
8:         $rating \leftarrow$ RateSuggestion($bestSuggestion$, LLM)
9:         Append $rating$ to $qualityRatings$
10:     **end for**
11:     $averageRating \leftarrow$ Average($qualityRatings$)
12:     **return** $averageRating$
13: **end procedure**
14: **function** GENERATESUGGESTIONS($\mathcal{G}^*$, $P$, LLM)
15:     **return** A set of suggestions for missing nodes based on $P$
16: **end function**
17: **function** SELECTBESTSUGGESTION($suggestions, v_{\text{GT}}$,LLM)
18:     Prompt LLM with $\mathcal{G}^*$, $v_{\text{GT}}$, and $suggestions$
19:     **return** LLM's choice of the best fitting suggestion
20: **end function**
21: **function** RATESUGGESTION($suggestion, LM$)
22:     Prompt LLM to rate $suggestion$ on a scale of 1 to 10
23:     **return** LLM's rating
24: **end function**

---

| **Ground Truth**: | Education up to high school or university degree |
|---|---|
| *Top ranked suggestion*: | Education level |
| Rating : | 9.5 |
| **Ground Truth**: | Pollution |
| *Top ranked suggestion*: | Smoking history |
| Rating : | 2.0 |
| **Ground Truth**: | Bonchitis |
| *Top ranked suggestion*: | smoking behavior |
| Rating : | 2.0 |
| **Ground Truth**: | Lung XRay report |
| *Top ranked suggestion*: | Lung Damage |
| Rating : | 8.0 |
| **Ground Truth**: | Socioeconomic status |
| *Top ranked suggestion*: | Driver's lifestyle |
| Rating : | 7.0 |

Table 6: Examples of model suggestions from and the corresponding LLM-as-judge score for a missing node variable.

**Shortcomings of LLM-as-judge.** LLM-as-judge uses GPT-4 as a judge model which could be biased towards some data. Since the training datasets are not public for this model, it would be hard

| |
|---|
| **Ground Truth**: Dyspnea laboured breathing |
| **LLM Suggestion**: Shortness of breath |
| Semantic similarity to GT: 0.57 |
| LLM-as-Judge score: 9.5 |

Table 7: Example comparing the semantic similarity and LLM-as-Judge metrics. Dyspnea is a medical term for shortness of breath. In this example, the contextual information, beyond exact matching, is better captured by LLM-as-Judge.

to judge how these biases might affect the final score. Hence for robust evaluation we also evaluate using the semantic similarity.

### A.6 ITERATIVELY HYPOTHESIZING IN OPEN WORLD

For each order, the algorithm prompts the LLM to generate mediator suggestions, selects the suggestion with the highest semantic similarity to the context, and iteratively updates the partial graph with these mediators. $\Delta$, quantifies the impact of mediator ordering by comparing the average highest semantic similarity scores obtained from both descending and ascending orders. This methodical evaluation sheds light on how the sequence in which mediators are considered might affect the LLM's ability to generate contextually relevant and accurate predictions.

---

**Algorithm 3** Random Order Mediator Hypothesis

---

1: **Input:** Partial graph $\mathcal{G}^*$ (where $\mathcal{G}^* = \mathcal{G} - H$), Treatment $v_t$, Outcome $v_y$, Number of mediators $H$, Number of suggestions $k$
2: **Output:** Updated graph $\mathcal{G}^*$ with selected mediators
3: **procedure** GENERATEMEDIATORSRANDOM($\mathcal{G}^*, v_t, v_y, H, k$)
4:     **for** $i \leftarrow 1$ to $H$ **do**
5:         $suggestions \leftarrow$ Generate $k$ suggestions for $v_{m_i}$ using $P_{\text{LLM}}(\mathcal{G}^*)$
6:         Initialize $highestSimilarity \leftarrow 0$
7:         Initialize $selectedMediator \leftarrow$ null
8:         **for** each $suggestion$ in $suggestions$ **do**
9:             $similarityScore \leftarrow$ Calculate semantic similarity for $suggestion$
10:             **if** $similarityScore > highestSimilarity$ **then**
11:                 $highestSimilarity \leftarrow similarityScore$
12:                 $selectedMediator \leftarrow suggestion$
13:             **end if**
14:         **end for**
15:         Update $\mathcal{G}^* \leftarrow \mathcal{G}^* \cup \{selectedMediator\}$
16:     **end for**
17:     **return** $\mathcal{G}^*$
18: **end procedure**

---

---

**Algorithm 4** Ordered Mediator Generation and Evaluation Based on MIS

---

1: **Input:** Partial graph $\mathcal{G}^*$, Treatment $v_t$, Outcome $v_y$, Set of potential mediators $M$, Number of suggestions $k$

2: **Output:** $\Delta$ - measure of the influence of mediator ordering

3: **procedure** CALCULATEMIS($v_t, v_y, M$)

4:      Initialize MISList as an empty list

5:      **for** each mediator $v_{m_i}$ in $M$ **do**

6:          Calculate NIE($v_{m_i}$) and NDE($v_{m_i}$)

7:          MIS($v_{m_i}$) $\leftarrow \frac{\text{NIE}(v_{m_i})}{\text{NDE}(v_{m_i})}$

8:          Append MIS($v_{m_i}$) to MISList

9:      **end for**

10:      **return** MISList

11: **end procedure**

12: **procedure** GENERATEMEDIATORSORDERED($\mathcal{G}^*, v_t, v_y, M, k$)

13:      MISList $\leftarrow$ CALCULATEMIS($v_t, v_y, M$)

14:      Sort $M$ in descending order of MISList to get $M_{\text{desc}}$

15:      Sort $M$ in ascending order of MISList to get $M_{\text{asc}}$

16:      $averageDesc \leftarrow$ GENERATEANDEVALUATE($\mathcal{G}^*, M_{\text{desc}}, k$)

17:      $averageAsc \leftarrow$ GENERATEANDEVALUATE($\mathcal{G}^*, M_{\text{asc}}, k$)

18:      $\Delta \leftarrow \frac{|averageDesc - averageAsc|}{averageDesc}$

19:      **return** $\Delta$

20: **end procedure**

21: **function** GENERATEANDEVALUATE($\mathcal{G}^*, M_{\text{order}}, k$)

22:      Initialize similarityScores as an empty list

23:      **for** each mediator $v_{m_i}$ in $M_{\text{order}}$ **do**

24:          Perform the same steps as in the refined random order mediator generation

25:          (Generate $k$ suggestions, select the most similar, update $\mathcal{G}^*$)

26:          Append the highest similarity score to similarityScores

27:      **end for**

28:      **return** Average of similarityScores

29: **end function**

---

## B  CONFOUNDERS

|        | Sachs | Alarm1 | Alarm2 | Ins1 | Ins2 | Ins3 | Ins4 | Ins5 | Ins6 | Ins7 |
|--------|-------|--------|--------|------|------|------|------|------|------|------|
| Zephyr | 0.12  | 0.37   | 0.29   | 0.45 | 0.49 | 0.37 | 0.29 | 0.33 | 0.46 | 0.73 |
| Mixtral| 0.89  | 0.54   | 0.57   | 0.57 | 1.0  | 0.32 | 0.23 | 0.38 | 0.28 | 1.0  |
| Neural | 0.34  | 0.27   | 0.28   | 0.42 | 0.47 | 0.34 | 0.48 | 0.48 | 0.38 | 0.48 |
| LLama  | 0.27  | 0.39   | 0.44   | 0.55 | 1.0  | 0.29 | 0.22 | 0.57 | 0.45 | 1.0  |
| Mistral| 0.23  | 0.62   | 0.46   | 0.58 | 1.0  | 0.28 | 0.28 | 0.28 | 0.28 | 1.0  |
| GPT-3.5| 0.34  | 0.39   | 0.48   | 0.48 | 1.0  | 0.58 | 0.20 | 0.48 | 0.47 | 1.0  |
| GPT-4  | 0.91  | 0.49   | 0.44   | 0.62 | 0.39 | 0.58 | 0.44 | 0.58 | 0.52 | 1.0  |

Table 8: Semantic similarity

|        | Sachs | Alarm1 | Alarm2 | Ins1 | Ins2 | Ins3 | Ins4 | Ins5 | Ins6 | Ins7 |
|--------|-------|--------|--------|------|------|------|------|------|------|------|
| Zephyr | 0.10  | 0.40   | 0.30   | 0.45 | 0.60 | 0.40 | 0.40 | 0.30 | 0.70 | 0.80 |
| Mixtral| 0.95  | 0.70   | 1.0    | 0.75 | 1.0  | 0.80 | 0.20 | 0.20 | 0.20 | 1.0  |
| Neural | 0.30  | 0.60   | 0.30   | 1.0  | 0.60 | 0.30 | 0.80 | 0.30 | 0.40 | 0.60 |
| LLama  | 0.20  | 0.50   | 0.44   | 0.40 | 1.0  | 0.50 | 0.20 | 0.70 | 0.45 | 1.0  |
| Mistral| 0.20  | 0.90   | 0.80   | 0.55 | 1.0  | 0.30 | 0.20 | 0.70 | 0.30 | 1.0  |
| GPT-3.5| 0.40  | 0.50   | 0.48   | 0.30 | 1.0  | 0.75 | 0.40 | 0.75 | 0.60 | 1.0  |
| GPT-4  | 0.95  | 0.65   | 0.80   | 0.60 | 0.70 | 0.80 | 0.85 | 0.80 | 0.75 | 1.0  |

Table 9: LLM judge

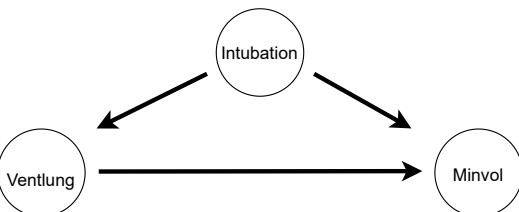

Figure 6: Alarm 1

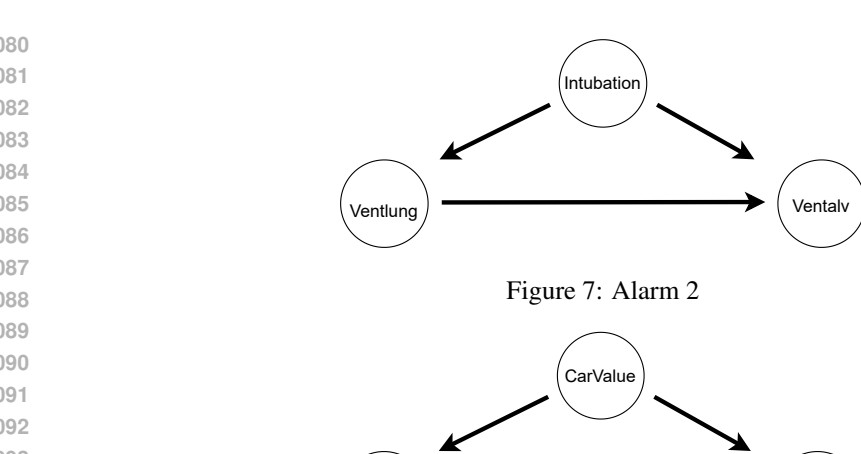

Figure 7: Alarm 2

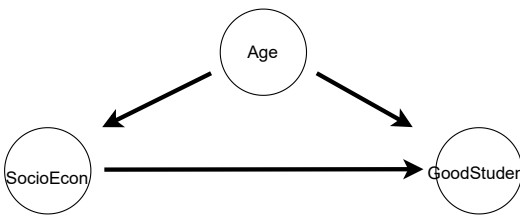

Figure 8: Insurance 1

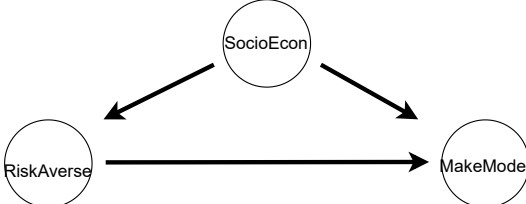

Figure 9: Insurance 2

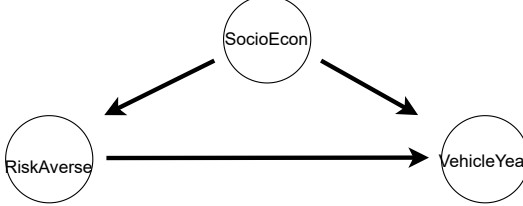

Figure 10: Insurance 3

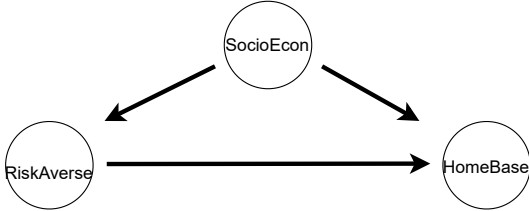

Figure 11: Insurance 4

Figure 12: Insurance 5

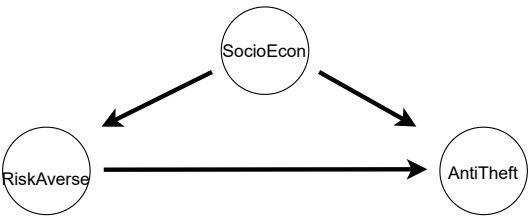

Figure 13: Insurance 6

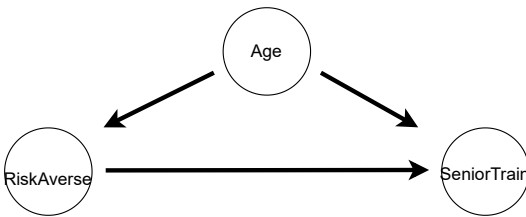

Figure 14: Insurance 7

## C FURTHER RESULTS

### C.1 VARIANCES

For brevity we didnt add variance in the main text, the following results have variances:

| | Cancer | | Survey | | Asia | | Alzheimers | | Child | | Insurance | | Alarm | | Avg | |
|---|---|---|---|---|---|---|---|---|---|---|---|---|---|---|---|---|
| | Sim | LLM-J | Sim | LLM-J | Sim | LLM-J | Sim | LLM-J | Sim | LLM-J | Sim | LLM-J | Sim | LLM-J | Sim | LLM-J |
| Zephyr | 0.36 ±0.04 | 0.61 ±0.06 | 0.34 ±0.07 | 0.60 ±0.05 | 0.45 ±0.05 | 0.66 ±0.04 | 0.35 ±0.03 | 0.75 ±0.03 | 0.51 ±0.02 | 0.70 ±0.04 | 0.45 ±0.04 | 0.44 ±0.05 | 0.46 ±0.03 | 0.69 ±0.02 | 0.42 ±0.04 | 0.63 ±0.04 |
| Mixtral | 0.41 ±0.03 | 0.66 ±0.04 | 0.39 ±0.05 | 0.66 ±0.06 | **0.66** ±0.02 | 0.75 ±0.03 | 0.31 ±0.04 | 0.77 ±0.02 | **0.53** ±0.03 | **0.77** ±0.02 | 0.46 ±0.03 | **0.56** ±0.04 | **0.50** ±0.03 | 0.72 ±0.06 | 0.46 ±0.03 | 0.70 ±0.05 |
| Neural | 0.38 ±0.02 | 0.77 ±0.05 | 0.43 ±0.02 | 0.55 ±0.03 | 0.53 ±0.03 | 0.55 ±0.04 | 0.44 ±0.05 | 0.71 ±0.03 | 0.48 ±0.04 | 0.70 ±0.03 | 0.47 ±0.04 | 0.43 ±0.05 | 0.47 ±0.03 | 0.67 ±0.04 | 0.45 ±0.03 | 0.63 ±0.04 |
| Llama | 0.40 ±0.03 | 0.48 ±0.05 | 0.40 ±0.04 | 0.54 ±0.05 | 0.53 ±0.03 | 0.58 ±0.06 | 0.45 ±0.05 | 0.61 ±0.03 | 0.48 ±0.04 | 0.63 ±0.03 | 0.42 ±0.01 | 0.34 ±0.05 | 0.46 ±0.02 | 0.65 ±0.03 | 0.45 ±0.03 | 0.55 ±0.04 |
| Mistral | 0.33 ±0.01 | 0.67 ±0.05 | 0.44 ±0.05 | 0.65 ±0.04 | 0.60 ±0.03 | 0.73 ±0.04 | 0.34 ±0.04 | 0.76 ±0.02 | 0.48 ±0.04 | 0.68 ±0.03 | 0.46 ±0.03 | 0.47 ±0.01 | 0.47 ±0.03 | 0.71 ±0.03 | 0.44 ±0.03 | 0.67 ±0.03 |
| GPT-3.5 | 0.48 ±0.03 | 0.74 ±0.04 | 0.42 ±0.00 | **0.79** ±0.03 | 0.47 ±0.04 | 0.61 ±0.04 | 0.39 ±0.05 | **1.00** ±0.00 | 0.36 ±0.05 | 0.60 ±0.05 | 0.47 ±0.07 | 0.52 ±0.02 | 0.48 ±0.04 | 0.73 ±0.05 | 0.44 ±0.04 | 0.71 ±0.03 |
| GPT-4 | **0.49** ±0.02 | **0.90** ±0.03 | **0.51** ±0.06 | 0.67 ±0.04 | **0.66** ±0.02 | **0.76** ±0.03 | **0.47** ±0.02 | 0.98 ±0.02 | 0.36 ±0.05 | 0.53 ±0.04 | **0.52** ±0.03 | **0.56** ±0.03 | 0.49 ±0.06 | **0.75** ±0.02 | **0.50** ±0.04 | **0.73** ±0.03 |

Table 10: Average semantic similarity and LLM-as-Judge metrics to evaluate LLMs in hypothesizing the missing variable in a causal DAG.

### C.2 ANALYSIS OF DIFFERENCE ACROSS TASKS

Since the metrics are different to evaluate each task, it is not meaningful or straightforward to compare the raw results. It must also be noted that the tasks are not linear. To address this, we rank the model performances across all models and datasets and present these rankings in Figure 15. This allows us to compare the relative performance of the models across different tasks.

As we observe from the graph, GPT-4 model shows consistently top performances in Tasks 1-3, however, it has one of the lowest performances for Task 4. GPT-3.5 shows a strong performance in Task 2 and 4, ranking 2nd, but drops in Tasks 1 and 3. We observe that Zephyr, Neural and Mistral show consistently average performances. These observations motivate the significance of the tasks proposed in our benchmark. They highlight the variability in model performance across different tasks and emphasize the need for comprehensive and diverse benchmarks to fully assess the capabilities of these models.

| | Asia | | Child | | Insurance | | Alarm | |
|---|---|---|---|---|---|---|---|---|
| | Sim | Δ | Sim | Δ | Sim | Δ | Sim | Δ |
| Zephyr | 0.61 ±0.03 | −0.02 ±0.01 | **0.54** ±0.04 | 0.17 ±0.02 | 0.47 ±0.05 | 0.19 ±0.02 | 0.51 ±0.05 | 0.20 ±0.02 |
| Mixtral | **0.87** ±0.02 | 0.01 ±0.01 | 0.50 ±0.05 | 0.18 ±0.02 | 0.48 ±0.05 | 0.15 ±0.02 | 0.52 ±0.05 | 0.13 ±0.01 |
| Neural | 0.65 ±0.06 | 0.04 ±0.02 | 0.48 ±0.05 | 0.21 ±0.02 | 0.42 ±0.04 | 0.16 ±0.02 | 0.46 ±0.04 | 0.12 ±0.01 |
| Llama | 0.80 ±0.08 | 0.07 ±0.02 | 0.49 ±0.05 | −0.05 ±0.01 | 0.44 ±0.06 | 0.21 ±0.02 | 0.51 ±0.05 | 0.07 ±0.01 |
| Mistral | 0.33 ±0.03 | 0.02 ±0.01 | 0.50 ±0.05 | 0.12 ±0.01 | 0.48 ±0.05 | 0.13 ±0.02 | 0.47 ±0.04 | 0.11 ±0.01 |
| GPT-3.5 | 0.48 ±0.05 | 0.01 ±0.01 | 0.36 ±0.04 | 0.25 ±0.03 | 0.48 ±0.05 | 0.17 ±0.02 | 0.51 ±0.05 | 0.02 ±0.01 |
| GPT-4 | 0.49 ±0.07 | 0.04 ±0.01 | 0.39 ±0.05 | 0.16 ±0.02 | **0.52** ±0.05 | 0.14 ±0.02 | **0.60** ±0.06 | −0.07 ±0.01 |

Table 11: Sim: semantic similarity for iteratively hypothesizing the mediator nodes when prompted with random order. Δ measures the change in the prediction of each model according to the MIS.

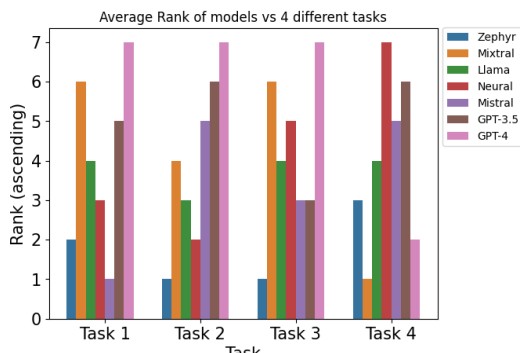

Figure 15: Average Rank of each model against the different tasks. We ranked the mode since the metrics are different to evaluate each task averaged across datasets

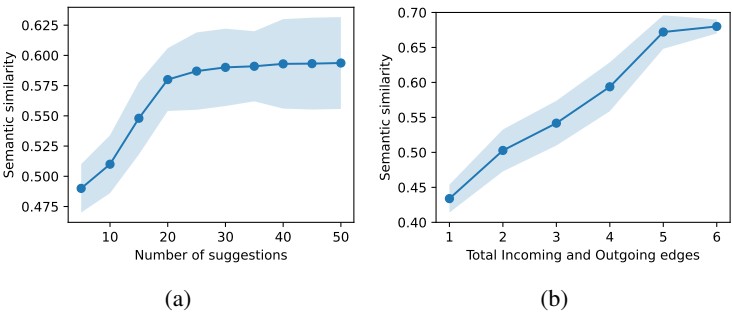

|       (a)       |       (b)       |

Figure 16: L: Plot of semantic similarity with an increasing number of suggestions for GPT-4 on the Alarm dataset. R: Plot of semantic similarity against the total number of incoming and outgoing edges for GPT-4 on the Alarm dataset.

## C.3 BREAKING DOWN THE PERFORMANCE

## C.4 EFFECT OF CONTEXT

We observed notable differences in the accuracy of LLM predictions for missing nodes within causal graphs when context was provided versus when it was absent. Specifically, the inclusion of contextual information about the causal graph significantly enhanced the LMs' ability to generate accurate and relevant predictions. In realistic settings, when this setup is being used by a scientist, they would provide the context of the task along with the partial graph. When context was not provided, the models often struggled to identify the most appropriate variables, leading to a decrease in prediction accuracy, especially for smaller models. Unsurprisingly, providing context was more important for smaller graphs than larger graphs. LLMs were able to understand the context of the graph via multiple other nodes in the graph for larger graphs.

|  | Cancer | | Survey | | Asia | | Insurance | | Alarm | |
|---|---|---|---|---|---|---|---|---|---|---|
|  | X | ✓ | X | ✓ | X | ✓ | X | ✓ | X | ✓ |
| In-Context | 0.75 | 1.00 | 0.67 | 1.00 | 0.68 | 0.88 | 0.85 | 0.90 | 0.96 | 0.96 |
| Out-of-Context | 0.00 | 0.25 | 0.33 | 0.33 | 0.53 | 0.61 | 0.58 | 0.58 | 0.60 | 0.57 |
| Open world Hypothesis | 0.39 | 0.41 | 0.40 | 0.39 | 0.63 | 0.66 | 0.49 | 0.50 | 0.44 | 0.46 |

Table 12: Model-Mixtral to evaluate the effect of context given in the prompt.

## C.5 USING EXPLANATIONS

While using LLMs for hypothesizing the missing nodes withing the causal graph for the open world setting, introduced an additional question to prompt the model to provide explanations for each of

their predictions. This was motivated by the fact that incorporating a rationale behind each prediction might enhance the model's semantic similarity. We present the results in the Table below: We observe that evaluating semantic similarity with explanations leads to a decrease in performance as compared to the earlier setting where the language model returned phrases. This is because semantic similarity, as a metric, evaluates the closeness of the model's predictions to the ground truth in a high-dimensional vector space, focusing on the semantic content encapsulated within the embeddings. It is a metric that leaves little room for interpretative flexibility, focusing strictly on the degree of semantic congruence between the predicted and actual variables. The introduction of explanations, while enriching the model's outputs with contextual insights, did not translate into improved semantic alignment with the ground truth.

| | Cancer | | Survey | | Asia | | Insurance | | Alarm | |
| | $X$ | ✓ | $X$ | ✓ | $X$ | ✓ | $X$ | ✓ | $X$ | ✓ |
|---|---|---|---|---|---|---|---|---|---|---|
| Sim | 0.49 ±0.02 | 0.38 ±0.07 | 0.51 ±0.06 | 0.44 ±0.10 | 0.66 ±0.02 | 0.57 ±0.09 | 0.52 ±0.03 | 0.40 ±0.07 | 0.49 ±0.06 | 0.40 ±0.06 |
| LLM-Judge | 0.90 ±0.03 | 0.91 ±0.02 | 0.67 ±0.04 | 0.69 ±0.02 | 0.76 ±0.03 | 0.76 ±0.04 | 0.56 ±0.03 | 0.55 ±0.03 | 0.75 ±0.02 | 0.75 ±0.02 |

Table 13: Model-GPT 4. Evaluating the effect of explanations on different metrics from Task 3.

**Ambiguous predictions which semantically represent the same variable.** An important linguistic concern that could be missed by semantic similarity is ambiguous hypothesis by the LLM that may have same semantics, which again breaks the semantic similarity metric. This further motivates LLM-judge metric whose input is - the context of the causal graph, the partial causal graph, the ground truth variable, and the model predictions. Given the rich context of the LLM-judge metric we suspect it would be able to overcome the ambiguity. We prompted the model to justify its hypothesis variables using explanations. We observe that evaluating semantic similarity with explanations leads to a decrease in performance as compared to the earlier setting where the language model returned just phrases. In Table 13 we observed a drop in performance for semantic similarity. In contrast, we observe a similar or slight improvement in the LLM-judge metric when the explanation of the model hypothesis is given.

## C.6 CHAIN OF THOUGHT

In recent times, Chain-of-Thought prompting has gained popularity due to its impressive performance in proving the quality of LLMs' output Kojima et al. (2022) also in metadata-based causal reasoning Vashishtha et al. (2023). We also incorporated COT prompting for our prompts. We perform ablation studies in Table. We observe that COT particularly improves the performance of the identification experiments.

| | Cancer | | Survey | | Asia | | Insurance | | Alarm | |
| | $X$ | ✓ | $X$ | ✓ | $X$ | ✓ | $X$ | ✓ | $X$ | ✓ |
|---|---|---|---|---|---|---|---|---|---|---|
| In-Context | 1.00 | 1.00 | 0.83 | 1.00 | 0.75 | 0.88 | 0.74 | 0.90 | 0.91 | 0.96 |
| Out-of-Context | 0.50 | 0.25 | 0.18 | 0.33 | 0.57 | 0.61 | 0.56 | 0.58 | 0.54 | 0.57 |

Table 14: Model-Mixtral to evaluate the effect of COT given in the prompt.

## C.7 ITERATIVE MEDIATOR SEARCH VS ALL AT ONCE

For Task 4, we iteratively hypothesize the missing variables (mediators). Our choice was primarily driven by the complexity of Task 4, which involves predicting multiple missing mediators, ranging from 1 to 10. For a Task with 10 missing mediators, the model would have to predict 50 suggestions at once. We initially hypothesized that LLMs might struggle with making multiple predictions across different variables simultaneously. This was indeed reflected in our results and GPT-4 outputs from Table X. The iterative approach allows the model's prediction to narrow the search space, which would not be possible in a non-iterative approach. This method is more aligned with the scientific discovery process, where hypotheses are often refined iteratively based on new findings. Furthermore,

our approach simulates a human-in-the-loop scenario, where the most plausible answer is selected and used to guide the next prediction.

|  | Asia | Child | Insurance | Alarm |
|---|---|---|---|---|
| Non-iterative | 0.42 +- 0.07 | 0.33 +- 0.06 | 0.45 +- 0.09 | 0.54 +- 0.05 |
| Iterative | 0.49 +- 0.05 | 0.39 +-0.03 | 0.52 +- 0.02 | 0.60 +- 0.04 |

## C.8 RESULTS ON NEUROPATHIC DATASET

We added a new dataset, the neuropathic pain dataset Tu et al. (2019), which is not part of common LLM training corpora as one needs to use a python script to download it. The dataset consists of 221 nodes and 770 edges, but for feasibility, we selected a subset of the graph for evaluation. We ran experiments for Task 1, Task 2, and Task 3.

| Model | Task 1 | Task 2 Result | Task 2 FNA | Task 3 Sim | Task 3 LLM-J |
|---|---|---|---|---|---|
| Mistral | 0.64 | 0.51 | 0.32 | 0.38 | 0.53 |
| Mixtral | 0.83 | 0.55 | 0.34 | 0.45 | 0.69 |
| Llama | 0.78 | 0.49 | 0.27 | 0.44 | 0.63 |
| GPT-3.5 | 0.82 | 0.53 | 0.31 | 0.47 | 0.72 |
| GPT-4 | 0.94 | 0.68 | 0.24 | 0.51 | 0.76 |

Table 15: Comparison of model performances across tasks on Neuropathic dataset.

## C.9 FINE GRAINED MODEL PERFORMANCE

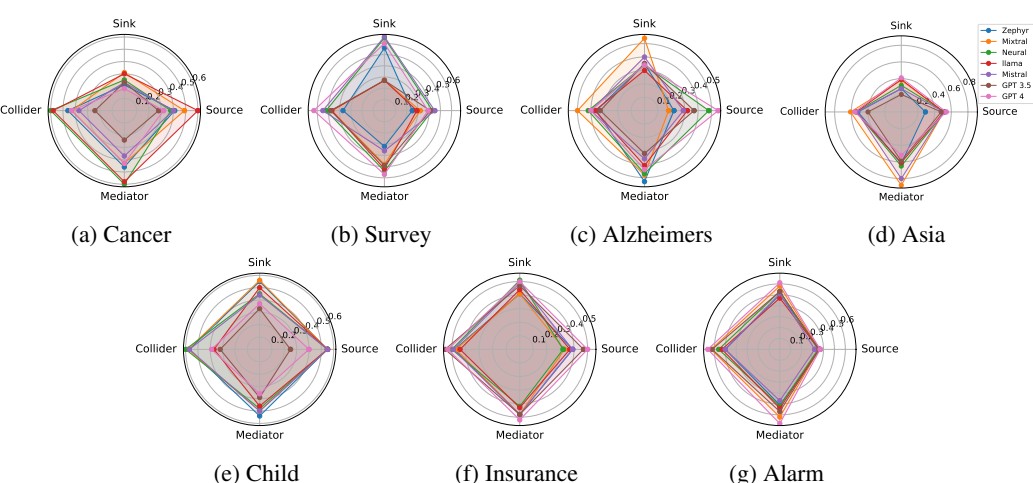

(a) Cancer  (b) Survey  (c) Alzheimers  (d) Asia

(e) Child  (f) Insurance  (g) Alarm

Figure 17: Detailed spider plots for Semantic similarity

# D  FINETUNING AND FEW-SHOT PROMPTING

## D.1  FINETUNING

we aim to assess the LLM's causal reasoning via prompting. Following are the reasons why fine-tuning is not the most practical solution:

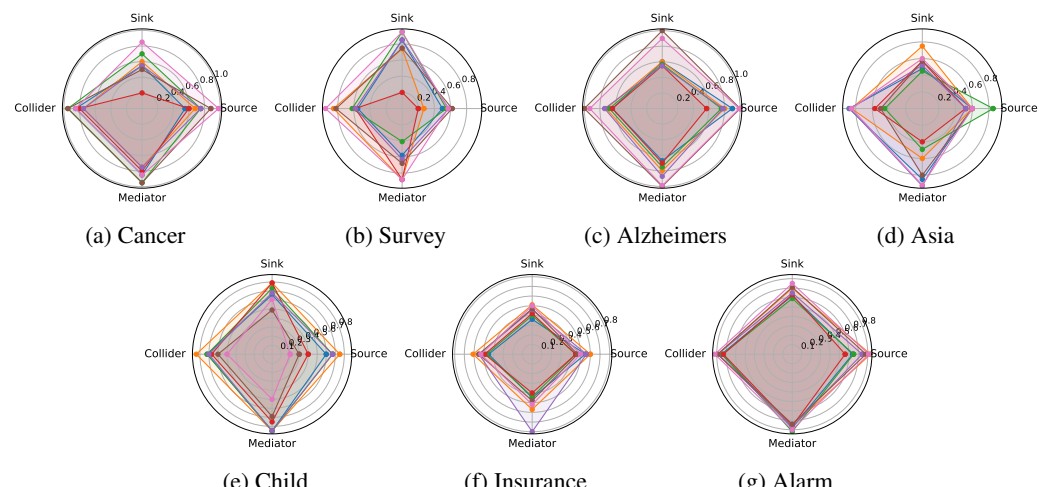

Figure 18: Detailed spider plots for LLM-as-judge metric

- Pretrained models come with a wealth of general knowledge, which we aim to leverage. Fine-tuning these models could potentially limit their ability to draw on this broad knowledge base. We aim to understand the utility of pretrained models, as fine-tuning large models like GPT-4 is not always feasible.
- The training dataset is too small for fine-tuning. Despite considering a large 52-edged graph: Insurance, we would have just 27 datapoints or Alarm with 37 datapoint. Additionally:

1. Using the same graph as part of train and test would unfortunately lead to training data leakage.
2. If we consider different graphs for train and test, there would exist a domain shift in the two graphs and the model may be overfitted to the domain of the train graph.

However, to illustrate our hypothesis and alleviate the reviewer's concern, we performed Supervised Fine-Tuning using QLoRA on the Mistral-7b-Instruct model for hypothesizing in the open world task. The train set here is all of the graphs minus the respective graph it was tested on. We tested on Survey, Insurance and Alzheimers graphs. The model was trained to give one best-fit suggestion for the missing variable.

|  | Insurance | Survey | Alzheimers |
|---|---|---|---|
| No fine-tuning | 0.42 +- 0.03 | 0.44 +- 0.05 | 0.34 +- 0.04 |
| Fine-tuned | 0.39 +- 0.04 | 0.39 +- 0.03 | 0.36 +- 0.07 |

Table 16: Finetuning results.

From the above results, it is evident that finetuning does not significantly improve over the prompting results. This is because during training the LLM gets biased towards the domains of training datasets which are contextually distant from the test domain, given the diversity of datasets chosen. One may think that training might help the LLM to understand the task, but from prompt-based model output, it was evident that the LLM can instruction-follow. In summary, we were able to extract the LLM knowledge via prompting and domain-specific fine-tuning could be closely looked at in the future works.

## D.2 FEWSHOT PROMPTING

Similar to fine-tuning, few-shot learning's success depends on balancing domain specificity and generality. To avoid test examples becoming part of the shots, we have to use different domains as

examples. Given the complexity of the Alarm graph, we decided to use them as a prior. We performed experiments with 1-shot and 5-shots for the Mixtral 8x7b model. We would like to remind you that

| Dataset | 0-shot | 1-shot | 5-shot |
|---|---|---|---|
| Cancer | 0.41 | 0.43 | 0.46 |
| Survey | 0.39 | 0.38 | 0.36 |
| Asia | 0.66 | 0.70 | 0.72 |
| Alzheimer's | 0.31 | 0.33 | 0.34 |
| Child | 0.53 | 0.55 | 0.56 |
| Insurance | 0.46 | 0.42 | 0.45 |

Table 17: Fewshot prompting results.

Alarm was a medical dataset which means that providing more examples in a different domain might hinder the model performance. Drop in performance when changing domain for in-context learning has been discussed in Kwan et al. (2024) and Gupta et al. (2024).

# E    CAUSAL GRAPHS

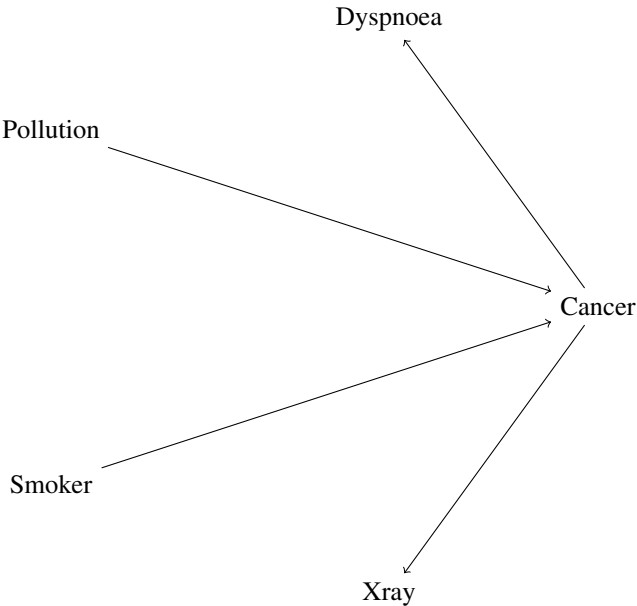

Figure 19: Cancer DAG

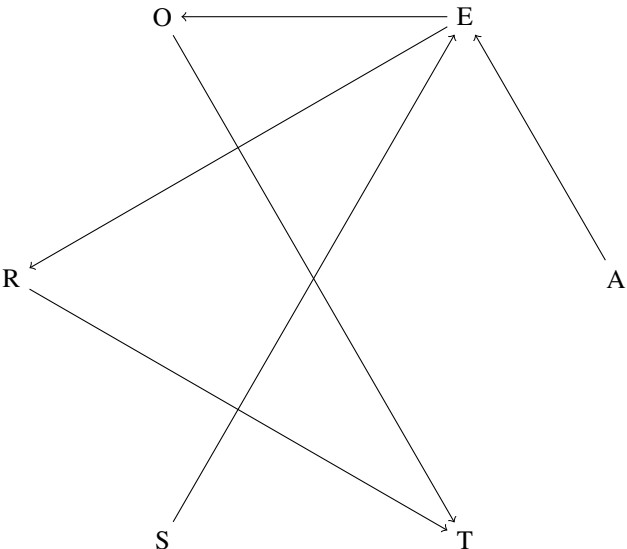

Figure 20: Survey DAG

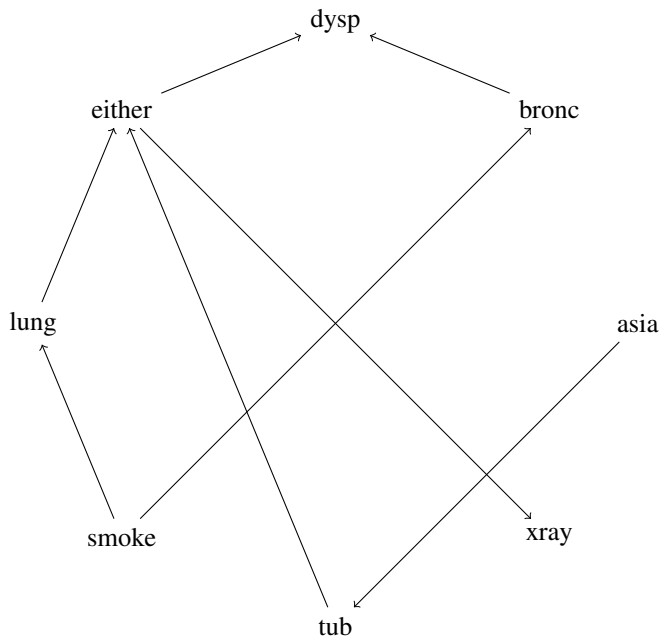

Figure 21: Asia DAG

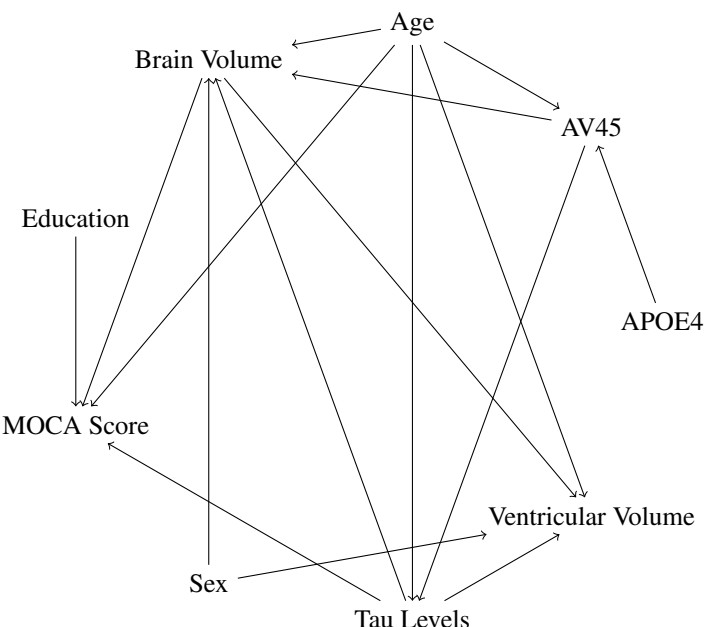

Figure 22: Alzheimer's DAG

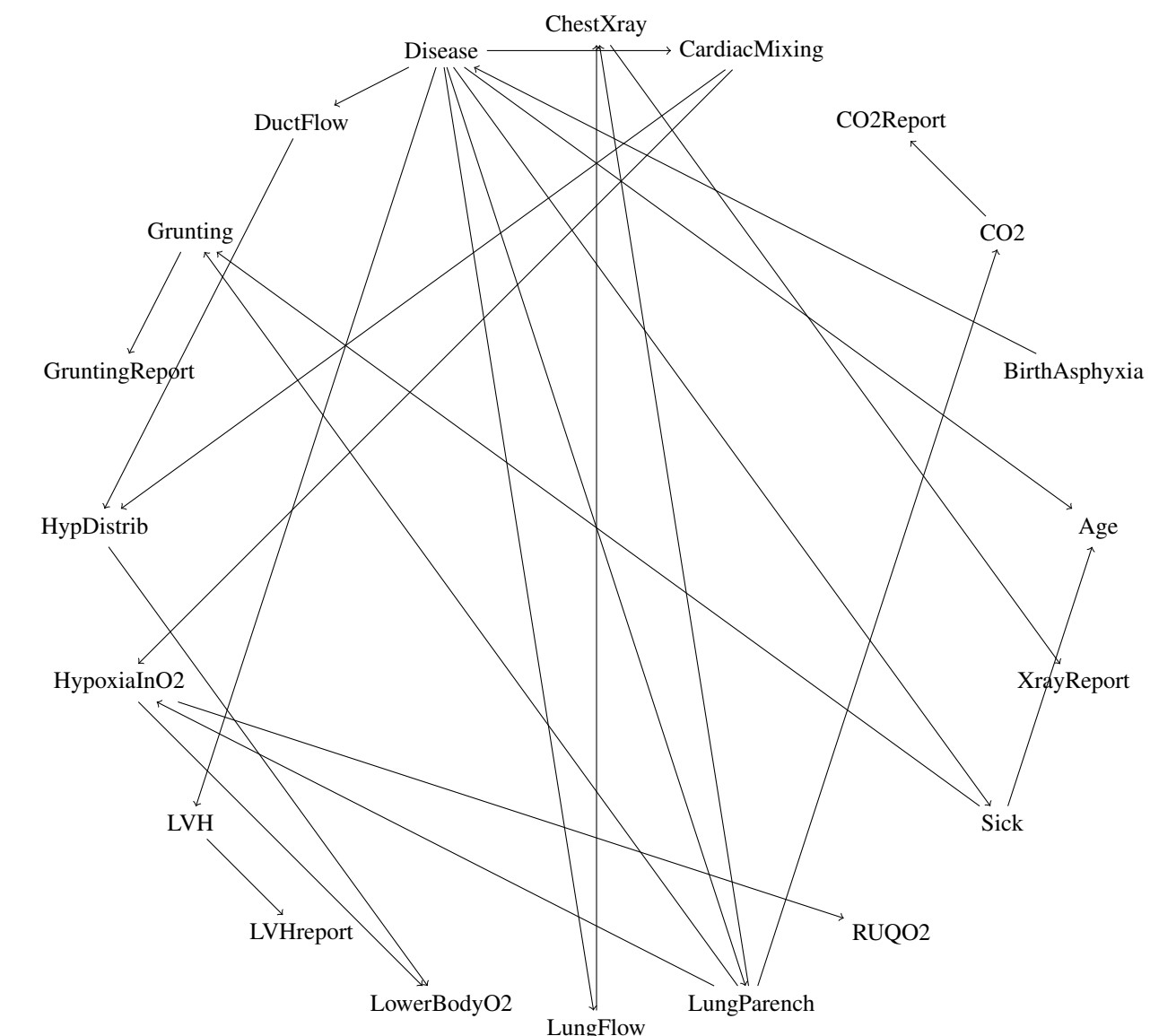

Figure 23: Child DAG

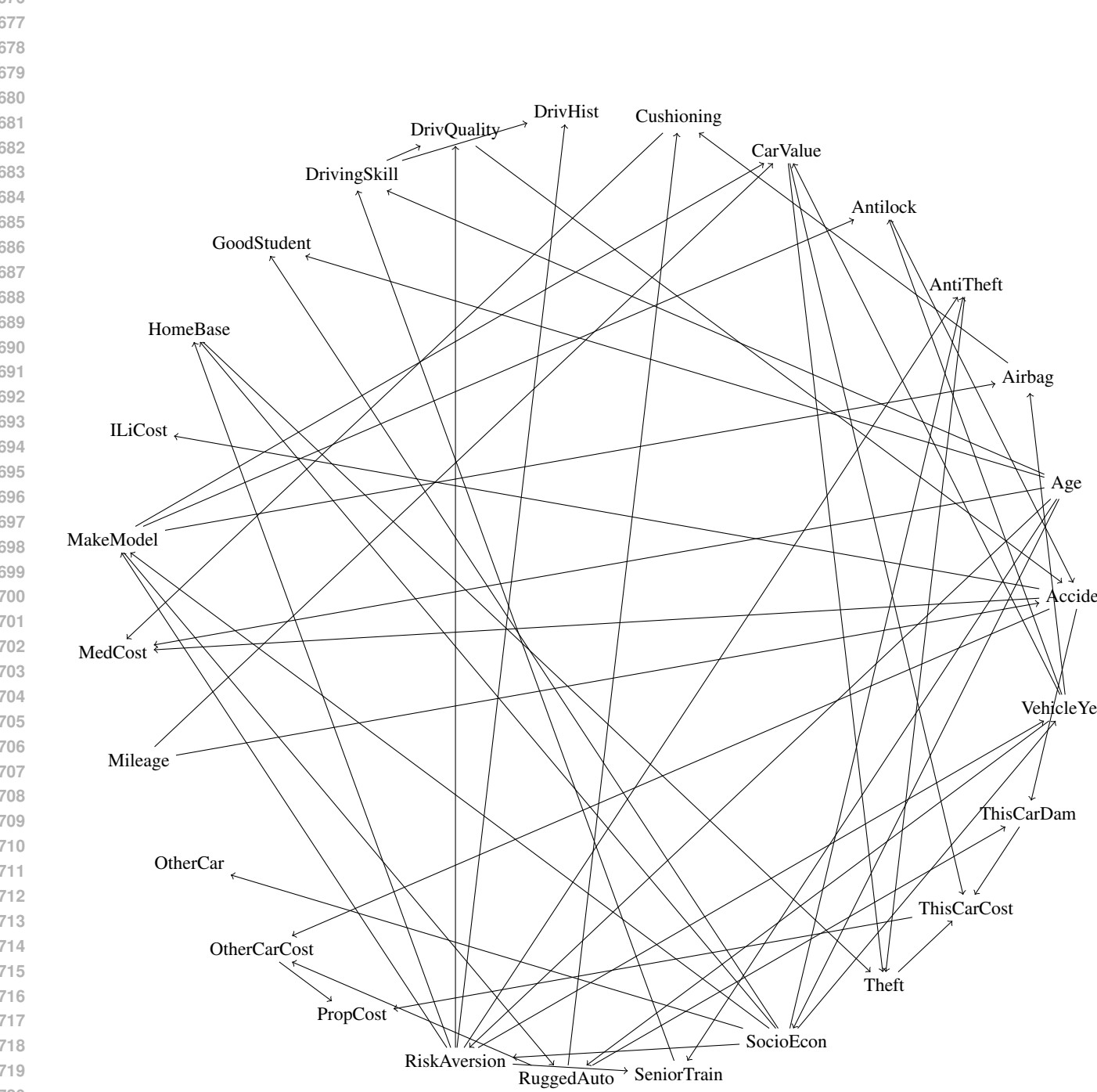

Figure 24: Insurance DAG

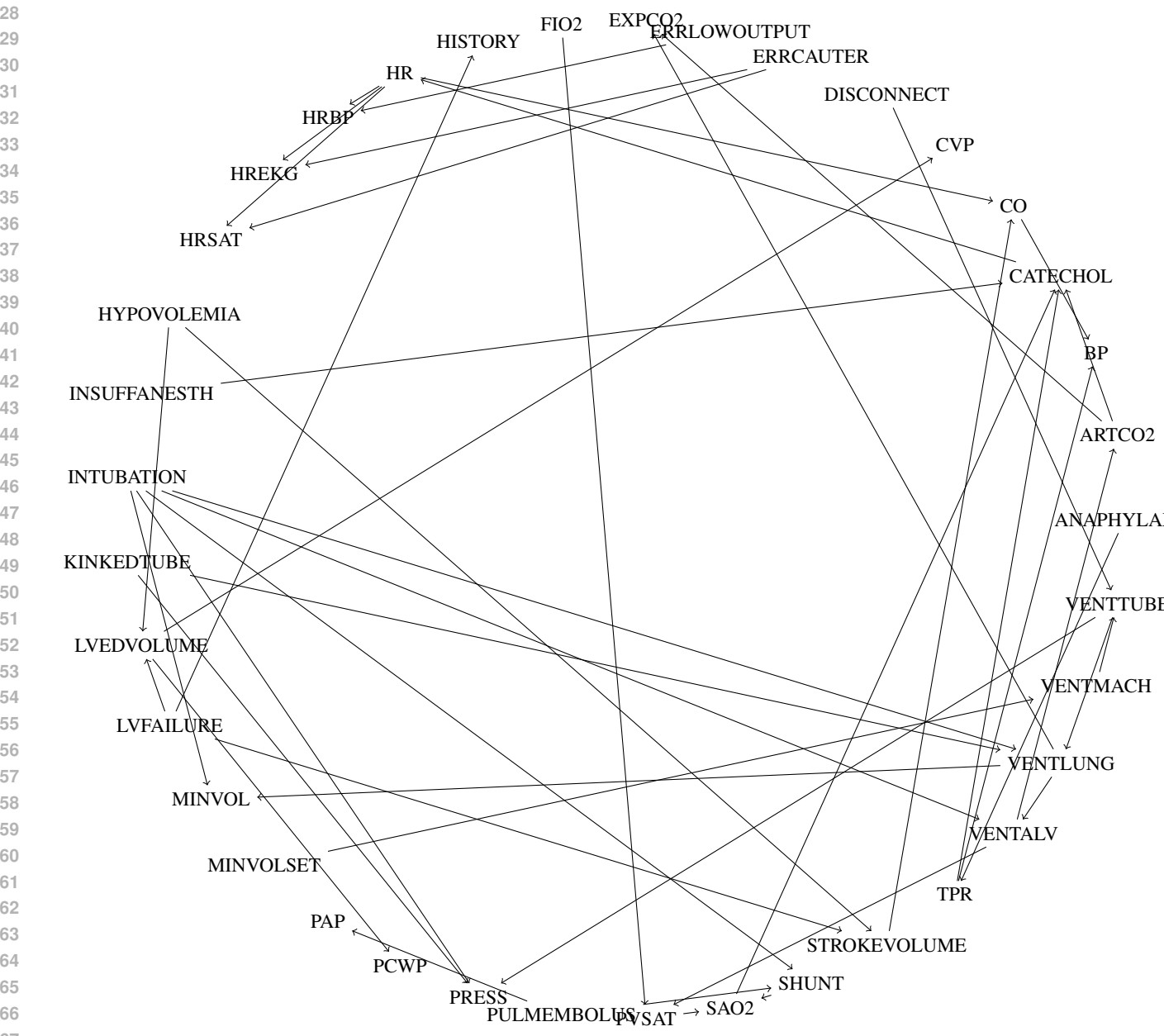

Figure 25: Alarm DAG

# F  PROMPT TEMPLATE

Hello. You will be given a causal graph. The context of the graph [CONTEXT]. Please understand the causal relationships between the variables - [VERBALISED DAG].

Prompt 1: Base prompt to describe the causal graph

Hello. You will be given a causal graph. The context of the graph is hypothetical patient monitoring system in an intensive care unit (ICU). Please understand the causal relationships between the variables - < anaphylaxis > causes < total peripheral resistance >. < arterial co2 > causes < expelled co2 >. < arterial co2 > causes < catecholamine >. < catecholamine > causes < heart rate >. < cardiac output > causes < blood pressure >. < disconnection > causes < breathing tube >. < error cauter > causes < heart rate displayed on ekg monitor >. < error cauter > causes < oxygen saturation >. < error low output > causes < heart rate blood pressure >. < high concentration of oxygen in the gas mixture > causes < pulmonary artery oxygen saturation >. < heart rate > causes < heart rate blood pressure >. < heart rate > causes < heart rate displayed on ekg monitor >. < heart rate > causes < oxygen saturation >. < heart rate > causes < cardiac output >. < hypovolemia > causes < left ventricular end-diastolic volume >. < hypovolemia > causes < stroke volume >. < insufficient anesthesia > causes < catecholamine >. < intubation > causes < lung ventilation >. < intubation > causes < minute volume >. < intubation > causes < alveolar ventilation >. < intubation > causes < shunt - normal and high >. < intubation > causes < breathing pressure >. < kinked chest tube > causes < lung ventilation >. < kinked chest tube > causes < breathing pressure >. < left ventricular end-diastolic volume > causes < central venous pressure >. < left ventricular end-diastolic volume > causes < pulmonary capillary wedge pressure >. < left ventricular failure > causes < previous medical history >. < left ventricular failure > causes < left ventricular end-diastolic volume >. < left ventricular failure > causes < stroke volume >. < the amount of time using a breathing machine > causes < the intensity level of a breathing machine >. < sudden blockage in the pulmonary arteries > causes < shunt - normal and high >. < sudden blockage in the pulmonary arteries > causes < pulmonary artery pressure >. < pulmonary artery oxygen saturation > causes < oxygen saturation >. < oxygen saturation > causes < catecholamine >. < shunt - normal and high > causes < oxygen saturation >. < stroke volume > causes < cardiac output >. < total peripheral resistance > causes < catecholamine >. < total peripheral resistance > causes < blood pressure >. < alveolar ventilation > causes < arterial co2 >. < alveolar ventilation > causes < pulmonary artery oxygen saturation >. < lung ventilation > causes < expelled co2 >. < lung ventilation > causes < minute volume >. < lung ventilation > causes < alveolar ventilation >. < the intensity level of a breathing machine > causes < breathing tube >. < breathing tube > causes < lung ventilation >. < breathing tube > causes < breathing pressure >.

Prompt 2: An example of the base prompt for Alarm dataset. Each relationship is enclosed in pointed brackets,<> followed by a full stop.

**Prompt:** Hello. You will be given a causal graph. The context of the graph is different aspects of a patient's health related to lung diseases. Please understand the causal relationships between the variables - < x > causes < tuberculosis >. < bronchitis > causes < dyspnoea-laboured breathing >. < individual has either tuberculosis or lung cancer > causes < positive xray >. < individual has either tuberculosis or lung cancer > causes < dyspnoea-laboured breathing >. < lung cancer > causes < individual has either tuberculosis or lung cancer >. < smoking cigarettes > causes < lung cancer >. < smoking cigarettes > causes < bronchitis >. < tuberculosis > causes < individual has either tuberculosis or lung cancer >.Based on your understanding and analytical skills, please identify what the variable X could most likely be from the given multiple choices. It's important to note that you MUST select an option from the provided choices, as one of them is the correct answer. The choices are: [ visited Asia; weather; book sales; movie rating ]. Please think step by step. Your answer should be formatted as follows - Answer: X = choice. The choice must be one of the provided options, and it should be formatted exactly as it appears in the choices. Generating your own options is not allowed in this task.

**Answer**: X = visited Asia.My thought process is as follows:The causal graph suggests that X causes tuberculosis, and tuberculosis is related to lung diseases. Visiting Asia is a possible explanation for why an individual may have been exposed to tuberculosis, as the disease is more prevalent in some Asian countries. Therefore, visited Asia is the most likely option for X.

Prompt 3: Out-of-context controlled variable identification, Ground truth variable: visited Asia

1890
1891
1892
1893
1894
1895
1896
1897
1898
1899
1900
1901
1902
1903
1904
1905
1906
1907
1908
1909
1910
1911
1912
1913
1914
1915
1916
1917
1918
1919
1920
1921
1922
1923
1924
1925
1926
1927
1928
1929
1930
1931
1932
1933
1934
1935
1936
1937
1938
1939
1940
1941
1942
1943

**Prompt:** Hello. You will be given a causal graph. The context of the graph is different aspects of a patient's health related to lung diseases Please understand the causal relationships between the variables - < x > causes < tuberculosis >. < bronchitis > causes < dyspnoea-laboured breathing >. < individual has either tuberculosis or lung cancer > causes < y >. < individual has either tuberculosis or lung cancer > causes < dyspnoea-laboured breathing >. < lung cancer > causes < individual has either tuberculosis or lung cancer >. < smoking cigarettes > causes < lung cancer >. < smoking cigarettes > causes < bronchitis >. < tuberculosis > causes < individual has either tuberculosis or lung cancer >. Based on your understanding and analytical skills, please identify what the variable X could most likely be from the given multiple choices. It's important to note that you MUST select an option from the provided choices, as one of them is the correct answer. The choices are: [ weather; book sales; movie rating; positive xray; visited Asia ]. Please think step by step. Your answer should be formatted as follows - Answer: X = choice. The choice must be one of the provided options, and it should be formatted exactly as it appears in the choices. Generating your own options is not allowed in this task.

**Answer**: Answer: X = visited Asia. My thought process is as follows:1. The graph shows that X causes tuberculosis, which suggests that X is a variable related to healthcare or medicine. 2. The graph also shows that bronchitis causes dyspnoea-laboured breathing, which further supports the idea that X is a variable related to healthcare or medicine. 3. The graph shows that individual has either tuberculosis or lung cancer causes y, which suggests that X could be a variable related to diagnosing or treating lung diseases. 4. The graph shows that smoking cigarettes causes lung cancer and bronchitis, which suggests that X could be a variable related to smoking or lung health. 5. Based on the above observations, it seems unlikely that X would be weather, book sales, or movie rating.6. Visited Asia is a variable that could potentially relate to healthcare or medicine, particularly if the patient has traveled to a region with a high prevalence of tuberculosis or other lung diseases.7. Therefore, based on the information provided in the graph, it seems most likely that X = visited Asia.

Prompt 4: In-context controlled variable identification, Ground truth variable: visited asia

**Prompt:** Hello. You will be given a causal graph. The context of the graph is different aspects of a patient's health related to lung diseases. Please understand the causal relationships between the variables - < x > causes < tuberculosis >. < bronchitis > causes < dyspnoea-laboured breathing >. < individual has either tuberculosis or lung cancer > causes < positive xray >. < individual has either tuberculosis or lung cancer > causes < dyspnoea-laboured breathing >. < lung cancer > causes < individual has either tuberculosis or lung cancer >. < smoking cigarettes > causes < lung cancer >. < smoking cigarettes > causes < bronchitis >. < tuberculosis > causes < individual has either tuberculosis or lung cancer >. Using your causal knowledge, can you suggest what variable X most likely? Give 5 suggestions for what variable X could be. Give reason for your suggestions. Finally, return your answer (without reasoning) in the following format: <Answer> [first suggestion, second suggestion, third suggestion .. and so on] </Answer> No explanations, no reasoning, just answers as a list as described earlier. For example, if the suggestions are A,B,C,D,E then - <Answer> [A, B, C, D, E] </Answer>

**Answer**: <Answer> [HIV/AIDS, Chronic respiratory conditions, **Pneumonia**, Exposure to asbestos, Silicosis] </Answer>
Reasons:
1. HIV/AIDS: It is known to weaken the immune system, making the individual more susceptible to infections like tuberculosis.
2. Chronic respiratory conditions: These can make an individual more susceptible to lung diseases like tuberculosis.
3. **Pneumonia**: It is a lung infection that can lead to serious conditions like tuberculosis if not treated properly.
4. Exposure to asbestos: Long-term exposure to asbestos can lead to lung diseases including lung cancer and tuberculosis.
5. Silicosis: It is a lung disease caused by inhalation of silica dust, leading to inflammation and scarring of the lung tissue, making the individual more susceptible to tuberculosis.

Prompt 5: Hypothesizing missing variable in open world, Ground truth variable: Visted Asia

**Prompt:** Hello. You will be given a causal graph. The context of the graph is different aspects of a patient's health related to lung diseases. Please understand the causal relationships between the variables - < visited asia > causes < tuberculosis >. < bronchitis > causes < dyspnoea-laboured breathing >. < individual has either tuberculosis or lung cancer > causes < positive xray >. < individual has either tuberculosis or lung cancer > causes < dyspnoea-laboured breathing >. < x > causes < individual has either tuberculosis or lung cancer >. < smoking cigarettes > causes < x >. < smoking cigarettes > causes < bronchitis >. < tuberculosis > causes < individual has either tuberculosis or lung cancer >. Using your causal knowledge, can you suggest what variable X most likely? Give 5 suggestions for what variable X could be. Give reason for your suggestions. Finally, return your answer (without reasoning) in the following format: <Answer> [first suggestion, second suggestion, third suggestion .. and so on] </Answer> No explanations, no reasoning, just answers as a list as described earlier. For example, if the suggestions are A,B,C,D,E then - <Answer> [A, B, C, D, E] </Answer>

**Answer**: <Answer> [**Lung Cancer**, Chronic Obstructive Pulmonary Disease, Emphysema, Pneumonia, Asthma] </Answer>
Reasons:
1. **Lung Cancer**: Smoking is a major cause of lung cancer and lung cancer can lead to tuberculosis or other lung diseases.
2. Chronic Obstructive Pulmonary Disease: Smoking is a leading cause of COPD and it can lead to other lung diseases including tuberculosis and lung cancer.
3. Emphysema: It is a type of COPD often caused by smoking and can lead to other lung diseases.
4. Pneumonia: Smoking increases the risk of pneumonia and pneumonia can lead to other lung diseases.
5. Asthma: Smoking can trigger asthma attacks and severe asthma can lead to other lung diseases.

Prompt 6: Hypothesizing missing variable in open world, Ground truth variable: Lung cancer

**Prompt:** Hello. You will be given a causal graph. The context of the graph is hypothetical patient monitoring system in an intensive care unit (ICU) Please understand the causal relationships between the variables - < anaphylaxis > causes < total peripheral resistance >. < x > causes < expelled co2 >. < x > causes < x3 >. < x3 > causes < x2 >. < x1 > causes < blood pressure >. < disconnection > causes < x7 >. < error cauter > causes < heart rate displayed on ekg monitor >. < error cauter > causes < oxygen saturation >. < error low output > causes < heart rate blood pressure >. < high concentration of oxygen in the gas mixture > causes < x9 >. < x2 > causes < heart rate blood pressure >. < x2 > causes < heart rate displayed on ekg monitor >. < x2 > causes < oxygen saturation >. < x2 > causes < x1 >. < hypovolemia > causes < left ventricular end-diastolic volume >. < hypovolemia > causes < stroke volume >. < insufficient anesthesia > causes < x3 >. < intubation > causes < x5 >. < intubation > causes < minute volume >. < intubation > causes < x4 >. < intubation > causes < shunt - normal and high >. < intubation > causes < breathing pressure >. < kinked chest tube > causes < x5 >. < kinked chest tube > causes < breathing pressure >. < left ventricular end-diastolic volume > causes < central venous pressure >. < left ventricular end-diastolic volume > causes < pulmonary capillary wedge pressure >. < left ventricular failure > causes < previous medical history >. < left ventricular failure > causes < left ventricular end-diastolic volume >. < left ventricular failure > causes < stroke volume >. < the amount of time using a breathing machine > causes < x6 >. < sudden blockage in the pulmonary arteries > causes < shunt - normal and high >. < sudden blockage in the pulmonary arteries > causes < pulmonary artery pressure >. < x9 > causes < x8 >. < x8 > causes < x3 >. < shunt - normal and high > causes < x8 >. < stroke volume > causes < x1 >. < total peripheral resistance > causes < x3 >. < total peripheral resistance > causes < blood pressure >. < x4 > causes < x >. < x4 > causes < x9 >. < x5 > causes < expelled co2 >. < x5 > causes < minute volume >. < x5 > causes < x4 >. < x6 > causes < x7 >. < x7 > causes < x5 >. < x7 > causes < breathing pressure >. Using your causal knowledge, can you suggest what variable X most likely? Give 5 suggestions for what variable X could be. Give reason for your suggestions. Finally, return your answer (without reasoning) in the following format: <Answer> [first suggestion, second suggestion, third suggestion .. and so on] </Answer> No explanations, no reasoning, just answers as a list as described earlier. For example, if the suggestions are A,B,C,D,E then - <Answer> [A, B, C, D, E] </Answer>

**Answer**: <Answer> [ventilation rate, respiratory effort, airway resistance, lung compliance, alveolar gas exchange] </Answer>

Reasons for the answer:

1. **Ventilation Rate**: The variable X causes both expelled CO2 and another variable (X3), which indirectly influences heart rate, blood pressure, and oxygen saturation. The rate of ventilation directly affects the amount of CO2 expelled from the body, making it a plausible candidate for X.

2. **Respiratory Effort**: This suggestion is based on the relationship between X and expelled CO2, as well as the influence of X on another variable (X3) that affects cardiovascular and respiratory parameters. Respiratory effort, which encompasses the work of breathing, could affect CO2 expulsion and have downstream effects on the cardiovascular system.

3. **Airway Resistance**: Airway resistance could be a candidate for X because it directly influences the amount of CO2 that can be expelled (a function of airflow) and could also affect other physiological variables through its impact on ventilation efficiency.

4. **Lung Compliance**: Lung compliance, or the ease with which the lungs can expand, directly affects ventilation and, consequently, the expulsion of CO2. It could also indirectly influence other variables in the graph through its impact on gas exchange efficiency.

5. **Alveolar Gas Exchange**: This process directly determines the amount of CO2 expelled and could influence the variable X3, affecting cardiovascular parameters. Efficient gas exchange is crucial for maintaining appropriate oxygen and CO2 levels in the blood, impacting heart rate, blood pressure, and oxygen saturation.

These suggestions are not explicitly mentioned in the input graph but are closely related to the physiological processes and outcomes described. They offer plausible explanations for the causal relationships involving the variable X, considering the context of patient monitoring in an ICU setting.

Prompt 7: Hypothesizing missing variable in open world #1 Ground truth variable: arterial CO2

## G  ASSUMPTIONS

The **causal sufficiency** of $\mathcal{G}$, by definition, implies that for every pair of variables within $\mathbf{V}$, all common causes are also included within $\mathbf{V}$. Extending this assumption to $\mathcal{G}^*$, we assume that the partial graph inherits causal sufficiency for its given that all edges among these variables are preserved as in $\mathcal{G}$. This preservation ensures that the observed relationships within $V^*$ are not confounded by omitted common causes. Since the faithfulness of $\mathcal{G}$ ensures that the observed conditional independencies among variables in $\mathbf{V}$ are accurately reflected by the causal structure represented by $\mathbf{E}$. By maintaining the same set of edges $\mathbf{E}$ in $\mathcal{G}^*$ for the subset $V^*$, we uphold the faithfulness assumption within the partial graph.

## H  NDE AND NIE

Average Treatment Effect (ATE) quantifies the expected change in the outcome $v_y$ caused by the unit change of the treatment $v_t$. ATE is part of the causal do-calculus introduced by Pearl (2009). We consider binary causal DAGs, i.e., each variable can either take 0 or 1 as values.

$$\text{ATE} = \mathbb{E}[v_y|\text{do}(v_t = 1)] - \mathbb{E}[v_y|\text{do}(v_t = 0)]$$

where the do($\cdot$) operator, represents an intervention. The $E[v_y|\text{do}(v_t = 1)]$ represents the expected value of the outcome variable $v_y$ when we intervene to set the treatment variable $v_t$ to 1 (i.e., apply the treatment), and $E[v_y|\text{do}(v_t = 0)]$ represents the expected value of $v_y$ when we set $v_t$ to 0 (i.e., do not apply the treatment).

### H.1  MEDIATION ANALYSIS

Mediation analysis is implemented to quantify the effect of a treatment on the outcome via a third variable, the mediator. The total mediation effect can be decomposed into the Natural Direct Effect (NDE) and the Natural Indirect Effect (NIE). The Natural Direct Effect (NDE) is the effect of the treatment on the outcome variable when not mediated by the mediator variable. The Natural Indirect Effect (NIE) is the effect of the treatment variable on the outcome variable when mediated by the mediator variable.

$$\text{NDE} = \mathbb{E}[v_{t=1}, v_{m=0} - v_{t=0}, v_{m=0}]$$

Here, NDE is calculated by comparing the expected outcome when the treatment variable is set to 1 and the mediator is fixed at the level it would take under the control treatment $v_t = 0$, with the expected outcome when both the treatment and the mediator are set to the control level.

$$\text{NIE} = \mathbb{E}[v_{t=0}, v_{m=1} - v_{t=0}, v_{m=0}]$$

Here, NIE is calculated by comparing the expected outcome when the treatment variable is set to 1 and the mediator is allowed to change as it would under the treatment, with the expected outcome when the treatment variable is set to 1 but the mediator is fixed at the control level.

**Prompt:** Strictly follow the format mentioned otherwise you will be disqualified.', 'ello. You will be given a causal graph. The context of the graph is hypothetical patient monitoring system in an intensive care unit (ICU) Please understand the causal relationships between the variables - < anaphylaxis > causes < total peripheral resistance >. < **Alveolar Gas Exchange** > causes < expelled co2 >. < **Alveolar Gas Exchange** > causes < x2 >. < x2 > causes < x1 >. < x > causes < blood pressure >. < disconnection > causes < x6 >. < error cauter > causes < heart rate displayed on ekg monitor >. < error cauter > causes < oxygen saturation >. < error low output > causes < heart rate blood pressure >. < high concentration of oxygen in the gas mixture > causes < x8 >. < x1 > causes < heart rate blood pressure >. < x1 > causes < heart rate displayed on ekg monitor >. < x1 > causes < oxygen saturation >. < x1 > causes < x >. < hypovolemia > causes < left ventricular end-diastolic volume >. < hypovolemia > causes < stroke volume >. < insufficient anesthesia > causes < x2 >. < intubation > causes < x4 >. < intubation > causes < minute volume >. < intubation > causes < x3 >. < intubation > causes < shunt - normal and high >. < intubation > causes < breathing pressure >. < kinked chest tube > causes < x4 >. < kinked chest tube > causes < breathing pressure >. < left ventricular end-diastolic volume > causes < central venous pressure >. < left ventricular end-diastolic volume > causes < pulmonary capillary wedge pressure >. < left ventricular failure > causes < previous medical history >. < left ventricular failure > causes < left ventricular end-diastolic volume >. < left ventricular failure > causes < stroke volume >. < the amount of time using a breathing machine > causes < x5 >. < sudden blockage in the pulmonary arteries > causes < shunt - normal and high >. < sudden blockage in the pulmonary arteries > causes < pulmonary artery pressure >. < x8 > causes < x7 >. < x7 > causes < x2 >. < shunt - normal and high > causes < x7 >. < stroke volume > causes < x >. < total peripheral resistance > causes < x2 >. < total peripheral resistance > causes < blood pressure >. < x3 > causes < **Alveolar Gas Exchange** >. < x3 > causes < x8 >. < x4 > causes < expelled co2 >. < x4 > causes < minute volume >. < x4 > causes < x3 >. < x5 > causes < x6 >. < x6 > causes < x4 >. < x6 > causes < breathing pressure >. Using your causal knowledge, can you suggest what variable X most likely? Give 5 suggestions for what variable X could be. Give reason for your suggestions. Finally, return your answer (without reasoning) in the following format: <Answer> [first suggestion, second suggestion, third suggestion .. and so on] </Answer> No explanations, no reasoning, just answers as a list as described earlier. For example, if the suggestions are A,B,C,D,E then - <Answer> [A, B, C, D, E] </Answer>

**Answer**: <Answer> [cardiac output, heart function, blood flow, vascular resistance, cardiac cycle] </Answer>
Reasons for the answer:
1. **Cardiac Output**: The variable X causes blood pressure and is caused by stroke volume. In physiological terms, cardiac output is a primary factor influencing blood pressure, as it represents the volume of blood the heart pumps per minute. Since stroke volume (the amount of blood pumped by the left ventricle of the heart in one contraction) contributes to cardiac output, it's logical to suggest "cardiac output" as a potential identity for X.
2. **Heart Function**: Given that X is related to blood pressure and influenced by stroke volume, it's reasonable to infer that X pertains to the overall function of the heart. Heart function encompasses various aspects of how the heart works, including pumping efficiency, which directly affects blood pressure.
3. **Blood Flow**: Since X has a direct relationship with blood pressure and an indirect relationship with factors affecting the heart's pumping ability, "blood flow" could be a plausible suggestion. Blood flow is crucial for maintaining blood pressure and is inherently linked to the heart's ability to pump blood effectively.
4. **Vascular Resistance**: X's impact on blood pressure and its connection to total peripheral resistance suggest that it might involve vascular resistance. Vascular resistance is the resistance that must be overcome to push blood through the circulatory system and is a key determinant of blood pressure.
5. **Cardiac Cycle**: The cardiac cycle involves all the events related to the flow of blood through the heart during one complete heartbeat, including systole (contraction) and diastole (relaxation). Given X's influence on blood pressure and its relationship with stroke volume, considering the cardiac cycle as a potential identity for X makes sense, as it encompasses the mechanisms affecting blood pressure regulation.

Prompt 8: Hypothesizing missing variable in open world #2 Ground truth variable: cardiac output

**Prompt:** Hello. You will be given a causal graph. The context of the graph is hypothetical patient monitoring system in an intensive care unit (ICU) Please understand the causal relationships between the variables - < anaphylaxis > causes < total peripheral resistance >. < < **Alveolar Gas Exchange** > causes < expelled co2 >. < < **Alveolar Gas Exchange** > causes < x1 >. < x1 > causes < x >. < **Cardiac Output** > causes < blood pressure >. < disconnection > causes < x5 >. < error cauter > causes < heart rate displayed on ekg monitor >. < error cauter > causes < oxygen saturation >. < error low output > causes < heart rate blood pressure >. < high concentration of oxygen in the gas mixture > causes < x7 >. < x > causes < heart rate blood pressure >. < x > causes < heart rate displayed on ekg monitor >. < x > causes < oxygen saturation >. < x > causes < **Cardiac Output** >. < hypovolemia > causes < left ventricular end-diastolic volume >. < hypovolemia > causes < stroke volume >. < insufficient anesthesia > causes < x1 >. < intubation > causes < x3 >. < intubation > causes < minute volume >. < intubation > causes < x2 >. < intubation > causes < shunt - normal and high >. < intubation > causes < breathing pressure >. < kinked chest tube > causes < x3 >. < kinked chest tube > causes < breathing pressure >. < left ventricular end-diastolic volume > causes < central venous pressure >. < left ventricular end-diastolic volume > causes < pulmonary capillary wedge pressure >. < left ventricular failure > causes < previous medical history >. < left ventricular failure > causes < left ventricular end-diastolic volume >. < left ventricular failure > causes < stroke volume >. < the amount of time using a breathing machine > causes < x4 >. < sudden blockage in the pulmonary arteries > causes < shunt - normal and high >. < sudden blockage in the pulmonary arteries > causes < pulmonary artery pressure >. < x7 > causes < x6 >. < x6 > causes < x1 >. < shunt - normal and high > causes < x6 >. < stroke volume > causes < **Cardiac Output** >. < total peripheral resistance > causes < x1 >. < total peripheral resistance > causes < blood pressure >. < x2 > causes < < **Alveolar Gas Exchange** >. < x2 > causes < x7 >. < x3 > causes < expelled co2 >. < x3 > causes < minute volume >. < x3 > causes < x2 >. < x4 > causes < x5 >. < x5 > causes < x3 >. < x5 > causes < breathing pressure > Using your causal knowledge, can you suggest what variable X most likely? Give 5 suggestions for what variable X could be. Give reason for your suggestions. Finally, return your answer (without reasoning) in the following format: <Answer> [first suggestion, second suggestion, third suggestion .. and so on] </Answer> No explanations, no reasoning, just answers as a list as described earlier. For example, if the suggestions are A,B,C,D,E then - <Answer> [A, B, C, D, E] </Answer>

**Answer**: <Answer> [patient anxiety level, patient consciousness level, patient pain level, patient respiratory rate, patient skin temperature] </Answer>
Reasons for the answer:
1. **Patient Anxiety Level**: The variable X influences heart rate, blood pressure, oxygen saturation, and cardiac output, which are all physiological parameters that can be affected by a patient's psychological state, such as anxiety. Anxiety can lead to increased heart rate and blood pressure, changes in oxygen saturation, and alterations in cardiac output due to the body's stress response.
2. **Patient Consciousness Level**: Consciousness level can impact heart rate, blood pressure, and oxygen saturation due to the brain's control over these vital signs. Changes in consciousness level, such as becoming more alert or lapsing into unconsciousness, can significantly affect these parameters, which are all influenced by variable X.
3. **Patient Pain Level**: Pain can cause physiological responses that include increased heart rate, elevated blood pressure, changes in oxygen saturation due to hyperventilation or altered breathing patterns, and potentially affect cardiac output as the body reacts to the stress of pain.
4. **Patient Respiratory Rate**: Given that X influences expelled CO2 and is affected by alveolar gas exchange, it's plausible that X could relate to the patient's respiratory rate. Respiratory rate directly impacts alveolar gas exchange, which in turn affects expelled CO2 levels, oxygen saturation, and indirectly influences heart rate and blood pressure through changes in blood chemistry.
5. **Patient Skin Temperature**: While not directly related to the cardiovascular or respiratory systems, skin temperature can be an indirect indicator of changes in these systems. For example, changes in cardiac output and peripheral resistance can affect blood flow to the skin, thereby influencing skin temperature. Variable X's impact on cardiac output and blood pressure suggests it could also indirectly affect skin temperature.
These suggestions are based on the understanding that variable X has a broad impact on both cardiovascular and respiratory parameters, indicating it could be a factor that influences or reflects the overall physiological state of the patient.

Prompt 9: Hypothesizing missing variable in open world #3 Ground truth variable: Heart rate

**Prompt:** Hello. You will be given a causal graph. The context of the graph is hypothetical patient monitoring system in an intensive care unit (ICU) Please understand the causal relationships between the variables - < anaphylaxis > causes < total peripheral resistance >. < < **Alveolar Gas Exchange** > causes < expelled co2 >. < < **Alveolar Gas Exchange** > causes < x >. < x > causes <Patient Respiratory Rate>. < **Cardiac Output** > causes < blood pressure >. < disconnection > causes < x4 >. < error cauter > causes < heart rate displayed on ekg monitor >. < error cauter > causes < oxygen saturation >. < error low output > causes < heart rate blood pressure >. < high concentration of oxygen in the gas mixture > causes < x6 >. <Patient Respiratory Rate> causes < heart rate blood pressure >. <Patient Respiratory Rate> causes < heart rate displayed on ekg monitor >. <Patient Respiratory Rate> causes < oxygen saturation >. <Patient Respiratory Rate> causes < **Cardiac Output** >. < hypovolemia > causes < left ventricular end-diastolic volume >. < hypovolemia > causes < stroke volume >. < insufficient anesthesia > causes < x >. < intubation > causes < x2 >. < intubation > causes < minute volume >. < intubation > causes < x1 >. < intubation > causes < shunt - normal and high >. < intubation > causes < breathing pressure >. < kinked chest tube > causes < x2 >. < kinked chest tube > causes < breathing pressure >. < left ventricular end-diastolic volume > causes < central venous pressure >. < left ventricular end-diastolic volume > causes < pulmonary capillary wedge pressure >. < left ventricular failure > causes < previous medical history >. < left ventricular failure > causes < left ventricular end-diastolic volume >. < left ventricular failure > causes < stroke volume >. < the amount of time using a breathing machine > causes < x3 >. < sudden blockage in the pulmonary arteries > causes < shunt - normal and high >. < sudden blockage in the pulmonary arteries > causes < pulmonary artery pressure >. < x6 > causes < x5 >. < x5 > causes < x >. < shunt - normal and high > causes < x5 >. < stroke volume > causes < **Cardiac Output** >. < total peripheral resistance > causes < x >. < total peripheral resistance > causes < blood pressure >. < x1 > causes < < **Alveolar Gas Exchange** >. < x2 > causes < x6 >. < x3 > causes < expelled co2 >. < x3 > causes < minute volume >. < x3 > causes < x1 >. < x3 > causes < x4 >. < x4 > causes < x3 >. < x4 > causes < breathing pressure > Using your causal knowledge, can you suggest what variable X most likely? Give 5 suggestions for what variable X could be. Give reason for your suggestions. Finally, return your answer (without reasoning) in the following format: <Answer> [first suggestion, second suggestion, third suggestion .. and so on] </Answer> No explanations, no reasoning, just answers as a list as described earlier. For example, if the suggestions are A,B,C,D,E then - <Answer> [A, B, C, D, E] </Answer>

**Answer**: <Answer> [oxygen delivery, arterial blood gas, lung compliance, blood oxygen level, respiratory efficiency] </Answer>

Reasons for the answer:

1. **Oxygen Delivery**: Given that "alveolar gas exchange" causes "X" and "X" in turn affects "patient respiratory rate," it's logical to infer that "X" could be related to how effectively oxygen is being delivered throughout the body. Oxygen delivery is a critical aspect of respiratory function and would directly impact respiratory rate.

2. **Arterial Blood Gas (ABG)**: ABG tests measure the levels of oxygen and carbon dioxide in the blood and can indicate how well oxygen is being transported and carbon dioxide is being expelled, which are functions influenced by alveolar gas exchange. Since "X" is caused by "alveolar gas exchange" and affects "patient respiratory rate," it's plausible that "X" could involve the measurement or state of arterial blood gases.

3. **Lung Compliance**: This refers to the elasticity or flexibility of the lungs, which directly impacts alveolar gas exchange and, consequently, the respiratory rate. Since "X" is linked to both "alveolar gas exchange" and "patient respiratory rate," it could suggest a relationship with the mechanical properties of the lung, such as compliance.

4. **Blood Oxygen Level**: This is a direct outcome of effective alveolar gas exchange and would naturally influence the patient's respiratory rate as the body adjusts to maintain adequate oxygenation. Given the causal links mentioned, "X" could very well represent the level of oxygen present in the blood.

5. **Respiratory Efficiency**: This encompasses how effectively the respiratory system can supply oxygen to and remove carbon dioxide from the body. It's influenced by alveolar gas exchange and directly affects the respiratory rate, making it a plausible candidate for "X."

These suggestions are not explicitly mentioned in the input graph but are logically inferred based on the described causal relationships and a basic understanding of respiratory physiology.

Prompt 10: Hypothesizing missing variable in open world #4 Ground truth variable: catecholamine. This repeats until all of the mediators are hypothesized.

