# OpenReview forum: "Filling the Gaps: LLMs for Causal Hypothesis Generation"
_ICLR.cc/2025/Conference — Submitted to ICLR 2025_

### Official Review · Reviewer_QiJv · 2024-10-18

**Soundness:** 2
**Presentation:** 3
**Contribution:** 2
**Rating:** 6
**Confidence:** 4

**Summary:**

The goal of this paper is to provide a benchmark for hypothesizing missing variables from a causal graph.

The paper designs progressively challenging scenarios
- partially known causal DAGs and a set of multiple choices for the missing variables.
- removing more than one node from the graph.
- hypothesize the missing variables of the causal DAGs without any explicit hints.

It formulates four tasks and has intensive experiments.

**Strengths:**

- This paper discusses an important and timely question. Making hypotheses on the missing causal variables is crucial to the scientific discovery process. A good benchmark is definitely needed.
- This paper provides clear definitions of its four tasks, like how each task is constructed and related.
- It conducts solid experiments with seven LLMs and multiple datasets. Code is provided in an anonymous repo.

**Weaknesses:**

The datasets used may already be included in the training corpus. The authors claim that the Alzheimer’s Disease dataset is new, and they cite an ICLR 2024 paper [1]. However, in section 4 (page 5) of [1], it is stated that the ground truth of Alzheimer’s Disease is discussed in [2] and [3], which are much earlier papers.


[1] Abdulaal, Ahmed, et al. "Causal Modelling Agents: Causal Graph Discovery through Synergising Metadata-and Data-driven Reasoning." The Twelfth International Conference on Learning Representations. 2023.

[2] Herrup, Karl. "The case for rejecting the amyloid cascade hypothesis." Nature neuroscience 18.6 (2015): 794-799.

[3] Walter Gulisano, Daniele Maugeri, Marian A Baltrons, Mauro F`a, Arianna Amato, Agostino Palmeri, Luciano D’Adamio, Claudio Grassi, DP Devanand, Lawrence S Honig, et al. Role of amyloid-β and tau proteins in Alzheimer’s disease: confuting the amyloid cascade. Journal of Alzheimer’s Disease, 64(s1):S611–S631, 2018.



Minors:
- Some notations should be revised, like ⊮in line 320.

**Questions:**

My main concern is the usage of the datasets. See the weakness part.

---

> ### Author Response · Authors · 2024-11-21
> **Response to the reviewer**
>
> We would like to thank the reviewer for their feedback on our submission. We answer the reviewers question below:
>
> >My main concern is the usage of the datasets. See the weakness part.
>
> Thank you for highlighting the concern regarding potential data leakage in our datasets. We have addressed this issue in our original submission (Lines 517–522). Paper [1] refers to papers [2] and [3] to acknowledge that the ground truth of Alzheimer’s Disease may be debatable. However, we considered the “consensus graph” from Figure 22 in [1], developed by the agreement of five domain experts. This consensus graph represents a carefully curated resource rather than relying solely on earlier assumptions in [2] and [3].
> Additionally, Table 5 in the Appendix provides examples of LLM-generated suggestions alongside their corresponding semantic similarity scores for missing node variables in Task 3. The non-verbatim nature of these completions indicates that the LLMs are not simply retrieving memorized patterns from their training data but generating novel hypotheses based on the provided prompts. This further supports the argument that data leakage concerns do not affect the models.
>
> To strengthen our argument, we have added a new dataset, the neuropathic pain dataset[a], which is not part of common LLM training corpora. The dataset consists of 221 nodes and 770 edges, but for feasibility, we selected a subset of the graph (22 nodes) for evaluation. We ran experiments for Task 1, Task 2, and Task 3. Below are the results:
>
> |         | Task 1 | Task 2(Result/FNA) | Task 3(Sim/LLM-J) |
> |---------|--------|--------------------|-----------------|
> | Mistral | 0.64   | 0.51/0.32          | 0.38/0.53       |
> | Mixtral | 0.83   | 0.55/0.34          | 0.45/0.69       |
> | Llama   | 0.78   | 0.49/0.27          | 0.44/0.63       |
> | GPT-3.5 | 0.82   | 0.53/0.31          | 0.47/0.72       |
> | GPT-4   | 0.94   | 0.68/0.24          | 0.51/0.76       |
>
>
> From the above, we observe that the results are consistent with the performance observed in other datasets. This consistency suggests that data leakage is not significantly impacting model performance. The dataset demonstrates the feasibility of the task in realistic settings and provides further evidence against data leakage concerns.
> We have included these results in the manuscript. We hope this alleviates the reviewer’s concern.
>
> We have added this discussion and results to the paper.
>
> [a] - Tu, Ruibo, et al. "Neuropathic pain diagnosis simulator for causal discovery algorithm evaluation." Advances in Neural Information Processing Systems 32 (2019).
>
> >Some notations should be revised, like ⊮in line 320.
>
> We have updated the paper.
>
> Thank you for your review. Your review has indeed improved our submission. We hope that our rebuttal has alleviated the reviewer's concerns about data leakage. Please do let us know if there are any remaining questions.

---

> > ### Comment · Reviewer_QiJv · 2024-11-23
> > **About the neuropathic pain dataset**
> >
> > Dear authors,
> >
> > Thanks for your effort on the additional experiment.
> >
> > The authors' rationale behind using [a], if I take it right, is that: the causal structure in [a] is learned from patient records, which are private, so it is unlikely to be accessed and fed into the pre-training process.
> >
> > However, I have to point out that the risk may still exist. For example, in figure 2 from [a], there is a link between "L C5-Radi" --> "L neck", so if one search "cervical radiculopathy Neck pain" in Google Scholar, there are many old papers stating their relations, I put some here:
> > - Eubanks, J. D. (2010). Cervical radiculopathy: nonoperative management of neck pain and radicular symptoms. American family physician, 81(1), 33-40.
> > - Rao, R. (2002). Neck pain, cervical radiculopathy, and cervical myelopathy: pathophysiology, natural history, and clinical evaluation. JBJS, 84(10), 1872-1881.
> >
> > Therefore, determining whether such a dataset is suitable for benchmarking may be out of the expertise of this reviewer. Here are my further suggestions:
> > - It is necessary to add a disclaimer about the potential issues.
> > - Characterize sufficient conditions of a dataset to produce a valid assessment under this paper's framework.
> > - Construct a simulated dataset carefully designed to avoid potential issues. (To the best of my knowledge, this is the strongest argument for avoiding data leakage. )
> >
> >
> >
> > I would like to keep my score. I acknowledge the motivation, but I am not sure whether the current submission is ready to provide a valid benchmark.

---

> ### Author Response · Authors · 2024-11-24
> **Response to the reviewer**
>
> We would like to thank the reviewer for their thoughtful response and for appreciating our experiment. We want to clarify the scope of our paper further.
>
> Consider a simple example: X causes lung cancer. From the reviewer's feedback, it seems they expect LLMs to avoid predicting smoking as a cause due to the extensive research on the topic. However, we would like to emphasize that it would be unreasonable to expect any system, including domain experts and LLMs, to propose something entirely outside of existing research when asked to hypothesize missing variables. Any reasonable hypothesis generation task relies on existing knowledge to propose plausible candidates, which is precisely the point of our framework (L58-65)
>
> - We would once again like to throw light on Table 5 in the Appendix provides examples of LLM-generated suggestions alongside their corresponding semantic similarity scores for missing node variables in Task 3. It is evident that there are no verbatim variable hypotheses generated. In the final version, we will include the entire list of suggestions with our codebase.
>
> - On the Disclaimer:
> We appreciate the suggestion of adding a disclaimer regarding data leakage. Our limitations section already addresses this concern, and we will further amplify it to make this point clearer in the manuscript.
>
> - On Sufficient Conditions for Dataset Validity:
> We would appreciate further clarification from the reviewer regarding the "sufficient conditions" they envision for our benchmark. We have already evaluated the model across nine different domains and are open to rephrasing or adjusting the graph encoding to present additional results. We are happy to revise and extend the dataset and task to address any remaining concerns.
>
> - On Simulated Datasets:
> Thank you for the suggestion to construct a simulated dataset. However, we suspect that any dataset we create might implicitly be part of pre-training data unless the graph diverges significantly from standard causal relationships. For example, replacing "smoking causes lung cancer" with "reading books causes lung cancer" may avoid direct pre-training issues, but it would be unrealistic for either a human or an LLM to hypothesize this as a plausible causal relationship. Additionally, it is important to recognize that causal graphs require supporting statistical data. Without such data, the graph would simply represent a correlation, which contradicts the principles of causal inference.
>
> We hope this response provides further clarity on the scope and limitations of our work and addresses the reviewer’s concerns. Thank you again for your valuable feedback.

---

> > ### Comment · Reviewer_QiJv · 2024-11-27
> >
> > The authors's response makes sense. Exposure to the existing literature seems necessary for LLMs and domain experts to give valuable answers, it can be difficult for LLMs to give entirely new variables. I agree with this. So here is my updated understanding on this paper:
> > - The paper expects LLMs to output some variables that have existed in the literature. This setting differs from my initial understanding, but still a reasonable one.
> > - The goal is to find some variables that are missing from a given causal graph but are in the literature. I am confused here. If this is the case, the benchmark is a knowledge retrieval one, and I failed to capture things related to making hypotheses. If one wants to test the ability to make hypotheses, one may need to avoid exposure to the literature about the missing nodes.
> >
> > Please correct me if I am wrong.

---

> > > ### Author Response · Authors · 2024-11-27
> > > **Response to the reviewer**
> > >
> > > Thank you for revisiting our response and for your updated understanding. We appreciate the opportunity to clarify further.
> > >
> > > We recognize that LLMs are unlikely to generate entirely novel variables absent from the literature, which is consistent with domain experts' behavior. The purpose of the benchmark is not to create altogether new scientific discoveries but to simulate the incremental process of hypothesis generation, where plausible, literature-supported candidates are proposed as a step toward guiding data collection or further investigation.
> > >
> > > The goal of our benchmark is not purely knowledge retrieval but to evaluate hypothesis generation constrained by causal reasoning. While LLMs may draw from their pre-trained knowledge, the task requires them to reason within the specific context of a partially defined causal graph. The LLM must propose plausible missing variables that satisfy the causal conditions imposed by the DAGs structure and relationships. This reasoning process ensures the generated hypotheses are not simple retrievals from literature but are grounded in the causal graph's context.
> > >
> > > We hope this helps?! Please do let us know if you have any questions or require further clarification.
> > >
> > > Best regards,
> > >
> > > Authors

---

> > > > ### Comment · Reviewer_QiJv · 2024-12-03
> > > >
> > > > Thanks for your further clarification.
> > > >
> > > > Simulating the incremental process of hypothesis generation makes sense. In this case, exposure to the previous related knowledge is helpful and necessary. The settings for task 3 and task 4 are good attempts. Therefore, I would like to increase the score to 6.
> > > >
> > > > The paper still has limitations. By causal graph, we mean the data-generating process of one specific system, e.g., some certain type of cells in its certain environment. The term "causal graph" used in this paper is more like a general one. For example, in Figure 19, we know in general, smoking can cause cancer. However, if we are studying people from a country where smoking is strictly forbidden, then this causal graph is not exactly the mechanism for this subpopulation.

---

> > > > > ### Author Response · Authors · 2024-12-03
> > > > >
> > > > > Thank you for your thoughtful and engaging discussion. We are glad that our framework aligns with your expectations.
> > > > >
> > > > > Regarding the limitation raised, we agree that causal graphs, as data-generating processes, can vary depending on the specific system or subpopulation under study. The generality of causal graphs in our benchmark reflects our goal to evaluate LLMs across diverse domains and contexts rather than focusing on highly specific or localized mechanisms. We acknowledge that subpopulation-specific graphs are an important consideration for real-world applications. To address this, we plan to discuss in the paper how the benchmark could be extended in future work to include context-specific causal graphs, enabling more nuanced evaluations tailored to particular systems or populations.
> > > > >
> > > > > Thank you again for your constructive suggestions, which will help improve the clarity and scope of our work. We would be happy to answer any further questions that may arise.

---

### Official Review · Reviewer_1syo · 2024-11-02

**Soundness:** 3
**Presentation:** 3
**Contribution:** 2
**Rating:** 6
**Confidence:** 4

**Summary:**

Authors propose new tasks that focus on the discovery of potential unobserved variables in a partially defined causal graph, hypothesising the true complete causal graph with the help of LLMs. Authors validated their approach against multiple LLMs, with different "difficulty levels" and prior knowledge about the causal graph.

**Strengths:**

- The tasks analysed by this work are relevant in the context of causality, especially in the field of causal discovery, where the complexity of the model scales significantly.
- Multiple LLMs are tested, providing a wide range of potential evidence in favor of the proposed approach.

**Weaknesses:**

- The multiple-choice question answering (MCQA) seems to be built beforehand, in such a way that the options are distant from the true one/ones w.r.t. the rest of the graph. This makes the evaluation easier.
- Sources/sinks are fundamentally different from mediators/confounders/colliders. The former expresses a property of the graph, while the latter expresses paths configurations. This makes the interpretation of the results unclear: I'm unsure if I should focus on the graph as a whole or in the paths of interest (if any).
- The evaluation is performed against "metrics" introduced by the authors themselves (e.g. MIS), without providing any property of the said metrics, making it harder to understand the actual the outcome of the evaluation.

**Questions:**

- Are "missing" variables actually "unobserved" variables? If yes, what is the choice behind this naming? Otherwise, what are the differences between these two types of variables?
- In causality the term "identification" refers to the identification of the causal estimand, hence, "causal identification" is misleading. Would it be possible to avoid this confusing overlap in the paper?
- Are the vertices (capital) V1 different from the vertices v1? Is this just a typo? Otherwise, what are the differences between these two types of vertices?
- Why Natural Direct Effect (NDE) and the Natural Indirect Effect (NIE) are taken as reference "metrics" for tasks evaluation?
- How are the NDE and NIE identified? The term "identified" here refers to the identification of the causal estimand, required for estimation of NDE/NIE.
- Why is Mediation Influence Score (MIS) a relevant metric in this context? How can we interpret it? What are the properties of such metric? Is it a metric at all? NIE and NDE are signed values, how to deal with sign reversal here?
- Can the authors provide a formal definition of "out-of-context distractors"? What is the difference between "in-context" and "out-of-context" choices? Aren't the names of the variables enough to provide context to the LLM? How is the context modeled?
- Authors say that: "The role of the LLM is to select a variable from the multiple choices". Apart from the true variable/variables in the list, how are the other multiple choices generated in the first place? In the appendix it is written that: "The words were randomly chosen to be far enough from the nodes", what does it mean "far enough"? Does this make the identification easier? What if we have variables that are indeed close to the true one/ones?

---

> ### Author Response · Authors · 2024-11-21
> **Response to the reviewer (1/3)**
>
> We would like to thank the reviewer for their detailed feedback. We answer the reviewer's questions and weaknesses below:
>
> > Q1: Are "missing" variables actually "unobserved" variables? If yes, what is the choice behind this naming? Otherwise, what are the differences between these two types of variables?
>
> In our paper, "missing" variables are not synonymous with "unobserved" variables in the traditional sense, such as latent variables or those hidden due to measurement error. Instead, they represent known unknowns within a partially defined causal graph. While missing variables can be considered a subset of unobserved variables, we intentionally use the term "missing" to align with the perspective of LLMs that see the variables as missing.
>
> > Q2: In causality the term "identification" refers to the identification of the causal estimand, hence, "causal identification" is misleading. Would it be possible to avoid this confusing overlap in the paper?
>
> We appreciate this observation and agree that the use of "causal identification" could be misleading. In the paper, "causal identification" refers to identifying missing variables within a causal DAG, which is distinct from the standard causal estimand identification.
> To avoid confusion, we propose renaming this to **missing variable identification**.
>
> > Q3: Are the vertices (capital) V1 different from the vertices v1? Is this just a typo? Otherwise, what are the differences between these two types of vertices?
>
> On Line 111, thank you for pointing out the typo, V1 is indeed v1. It is updated in the paper.
>
> > Q4: Why Natural Direct Effect (NDE) and the Natural Indirect Effect (NIE) are taken as reference "metrics" for tasks evaluation? How are the NDE and NIE identified? The term "identified" here refers to the identification of the causal estimand, required for estimation of NDE/NIE.
>
> NDE and NIE were used to evaluate the alignment between the LLM's hypothesized mediators and the statistical data from the DAGs, (only in) Task 4, which focuses on hypothesizing multiple mediators. These measures are well-established in causal inference for quantifying mediation effects, making them a natural choice to evaluate the task of mediator identification.
>
> The identification of NDE and NIE follows the do-calculus framework, leveraging the ground truth DAGs in the datasets to derive causal estimands. We recognize that NDE and NIE are not metrics in the conventional sense but rather components of mediation analysis. We acknowledge that NDE and NIE are not conventional metrics but are components of mediation analysis. Their inclusion in Task 4 provides a grounded way to evaluate whether the order of hypothesized mediators proposed by the LLM aligns with the statistical relationships in the data. This approach enhances the interpretability of the evaluation, as it directly links the performance of hypothesized mediators to their statistical data.
>
> > Q5: Why is Mediation Influence Score (MIS) a relevant metric in this context? How can we interpret it? What are the properties of such metric? Is it a metric at all? NIE and NDE are signed values, how to deal with sign reversal here?
>
> We would like to refer the reviewer to the answer to the previous question to describe the significance of the context. MIS evaluates the relative importance of hypothesized mediators by comparing the indirect mediation effect (NIE) to the direct effect (NDE). This makes it particularly relevant in contexts where prioritizing impactful mediators is crucial.
> - Interpretation: MIS provides a scale-free comparison of mediators, enabling prioritization based on their relative influence in the causal pathway.
> - Properties: MIS is always positive, as it evaluates the absolute contribution of mediators.
> Thank you for the suggestion. We have updated the submission with the above clarifications.

---

> ### Author Response · Authors · 2024-11-21
> **Response to the reviewer (2/3)**
>
> > Q6: Can the authors provide a formal definition of "out-of-context distractors"? What is the difference between "in-context" and "out-of-context" choices? Aren't the names of the variables enough to provide context to the LLM? How is the context modeled?
>
> As mentioned in Lines 205–206, out-of-context distractors are variables unrelated to the causal domain of the given DAG, chosen to minimize any contextual and overlap with the true missing variable. For example, if the DAG is the Asia graph, out-of-context distractors might include terms like "book sales" or "movie ratings." These variables are semantically distant from the causal relationships in the graph.
>
> In contrast, in-context distractors (introduced in Task 2) are plausible variables within the same causal graph. For example, for the Asia DAG, "tuberculosis" might serve as an in-context distractor when the true missing variable is "bronchitis."
> To clarify this distinction, Task 1 includes three out-of-context variables and one true missing variable, while Task 2 includes three out-of-context variables, one true missing variable, and one in-context distractor.
>
> We have updated the manuscript to include these details for better clarity.
>
> >Q7: Authors say that: "The role of the LLM is to select a variable from the multiple choices". Apart from the true variable/variables in the list, how are the other multiple choices generated in the first place? In the appendix it is written that: "The words were randomly chosen to be far enough from the nodes", what does it mean "far enough"? Does this make the identification easier? What if we have variables that are indeed close to the true one/ones?
>
> The distractors were generated by ensuring semantic dissimilarity from the DAG domain. "Far enough" refers to this semantic dissimilarity, operationalized by selecting words or variables unlikely to be relevant to the causal relationships in the DAG. For example, "book sales" would be considered semantically far from the Asia DAG.
>
> While this approach simplifies Task 1, it provides a baseline for assessing LLMs' causal reasoning. Task 2 includes in-context distractors to increase task complexity, requiring models to reason about plausible but incorrect options. This progression reflects a structured evaluation of LLM performance under varying difficulty levels. This simplifies Task 1 to establish a baseline for assessing LLMs' ability to identify missing variables. However, Task 2 introduces in-context distractors, which are semantically closer and domain-relevant but incorrect, making the task significantly more challenging. This progression ensures a structured evaluation of LLM performance under varying difficulty levels.
>
> By including both out-of-context and in-context distractors across tasks, the benchmark provides a comprehensive assessment of LLMs' causal hypothesis generation capabilities. We appreciate the reviewer’s observation and have further clarified this distinction in the manuscript.
>
> > W1: The multiple-choice question answering (MCQA) seems to be built beforehand, in such a way that the options are distant from the true one/ones w.r.t. the rest of the graph. This makes the evaluation easier.
>
> We might have missed the reference, but we would highly appreciate it if the reviewer could kindly point us to the reference with a similar setting to ours with multiple choices so that we can add it to our submission. For evaluation please refer to Q6 and Q7.
>
> > W2 : Sources/sinks are fundamentally different from mediators/confounders/colliders. The former expresses a property of the graph, while the latter expresses paths configurations. This makes the interpretation of the results unclear: I'm unsure if I should focus on the graph as a whole or on the paths of interest (if any).
>
> We agree with the reviewer that sources and sinks represent graph-level properties, while mediators, confounders, and colliders are path-specific configurations. Sources and sinks may primarily identify causal relationships involving treatments and outcomes. Mediators, confounders, and colliders: These nodes are evaluated in the context of specific causal pathways (e.g., backdoor or mediator paths) to assess their role in enabling or blocking causal effects. We present results on confounders, an important backdoor of study in causal inference speerated.
>
> To create a comprehensive benchmark, we aimed to model real-world scenarios where both structural (source/sink) and path-based (mediator/confounder/collider) hypotheses are essential. By doing so, the benchmark reflects the diverse task of causal hypothesis generation and allows for a broader evaluation of LLM capabilities.

---

> > ### Author Response · Authors · 2024-11-21
> > **Response to the reviewer (3/3)**
> >
> > > W3: The evaluation is performed against "metrics" introduced by the authors themselves (e.g. MIS), without providing any property of the said metrics, making it harder to understand the actual outcome of the evaluation.
> >
> > Please see our response for Q4 and Q5.
> >
> > We would like to kindly thank the reviewer for the great time and effort in reviewing our submission, The reviewer's comments have greatly improved our submission. Please do let us know if there are any remaining questions. If we have answered all of the reviewers' questions, we would appreciate it if they could kindly reconsider their score.

---

> ### Author Response · Authors · 2024-11-27
> **Any further questions?**
>
> Dear reviewer,
>
> Thank you for your insightful questions and suggestions. We have addressed your questions in detail and updated the paper. If we have adressed all of your concerns and questions, we would appreciate it, iit it could be reflected in the scores as well. We look forward to the opportunity to discuss any further questions.
>
> Best regards,
>
> Authors

---

> > ### Comment · Reviewer_1syo · 2024-12-02
> >
> > Dear authors,
> >
> > Thank you for the effort you made in improving the submission, I'll raise my score accordingly.
> >
> > Best regards,
> > Reviewer

---

> > > ### Author Response · Authors · 2024-12-02
> > >
> > > We want to thank the reviewer for reassessing our work. We are glad that our response and paper revision answered the reviewer's questions. We would be happy to answer any further questions.
> > >
> > > Best regards,
> > >
> > > Authors

---

### Official Review · Reviewer_oVmw · 2024-11-02

**Soundness:** 3
**Presentation:** 2
**Contribution:** 2
**Rating:** 5
**Confidence:** 4

**Summary:**

This paper proposes a novel benchmark designed to evaluate the ability of LLMs to hypothesize missing variables in causal directed acyclic graphs (DAGs). Motivated by the scientific discovery process, the objective is to simulate the generation of hypotheses in partially complete causal graphs, where LLMs hypothesize about missing nodes, particularly mediators, causes, and effects. The benchmark’s results indicate that LLMs do well at hypothesizing mediating variables between causes and effects, however, they struggle with hypothesizing direct causes and effects.

**Strengths:**

1. The paper presents a novel use of LLMs in scientific discovery by applying them to generate hypotheses through predicting causal variables in partially incomplete causal DAGs.

2. It provides a comprehensive evaluation of how various LLMs perform in predicting missing causal variables across different levels of difficulty. The benchmark assesses both open-source and proprietary LLMs, using diverse causal DAGs from multiple domains.

3. Given the recent focus on LLMs in scientific contexts, the benchmark is highly relevant. By introducing this task, the paper makes a valuable contribution to the community’s understanding of LLMs in scientific discovery, potentially inspiring future research to enhance LLMs’ causal reasoning abilities.

**Weaknesses:**

1. The paper could be clearer, especially in the methodology section, which does not adequately motivate each task or present them naturally. The tasks became clearer only upon reading the evaluation.

2. There is a risk of data leakage in most of the datasets used. While the authors attempt to address this limitation by presenting the causal graphs to the LLMs in a novel way if the LLMs were trained on the datasets, the underlying causal patterns could be memorized, and the benchmark’s validity could be compromised. This raises the need for further analysis to determine whether LLMs are genuinely performing causal reasoning or merely memorizing causal patterns.

3. The assumption that all causal relationships are known and correct may be unrealistic in practical applications. For example, it may be improbable to know the inbound and outbound edges of a mediator variable without knowing the variable itself. This restrictive assumption limits the benchmark’s ability to fully assess LLMs’ causal reasoning.

**Questions:**

1. In the paper, you argue that Task 3 represents a realistic scenario where the LLM must complete partial causal DAGs in an open-ended manner. While this setting makes it practical for a scalable benchmarking framework, is it actually realistic for users to create partially complete causal DAGs where the edges are known and correct? If all causal variables are not known, then is it practical to say the users can provide such a graph?

2. Is there a difference between Tables 3 and 11 reported in the paper? Both look almost the same. Additionally, can you clarify the metric ∆, since the captions in Tables 3 and 11 describe ∆ differently than what is described in Section 5.5? Finally, can you please explain why ∆ is better than simply presenting the difference in performance when mediators are iteratively presented to the LLM in ascending and descending orders of significance?

3. Section 4.2 is a bit unclear. Why are two variables removed, but the LLM is only required to predict one? Additionally, Figure 2b does not depict two variables being removed, so I am unsure why two are removed.

---

> ### Author Response · Authors · 2024-11-21
> **Response to the reviewer (1/2)**
>
> We would like to thank the reviewer for their valuable feedback. We answer their questions below:
>
> > Q1: In the paper, you argue that Task 3 represents a realistic scenario where the LLM must complete partial causal DAGs in an open-ended manner. While this setting makes it practical for a scalable benchmarking framework, is it realistic for users to create partially complete causal DAGs where the edges are known and correct? If all causal variables are not known, then is it practical to say the users can provide such a graph?
>
> We attempted to refrain from calling Task 3 a realistic setting. We consider that our benchmark serves as a proxy for the broader process of scientific discovery and hypothesis generation. In Section 5.4.1, we specifically investigate the model’s ability to hypothesize confounders without access to the existing edges in the DAG. We introduced the edges restriction to ensure a controlled and systematic evaluation of the LLM’s performance. Without this restriction, it would be challenging to evaluate the model's predictions as the search space for the hypotheses would be too large.
>
> > Q2: Is there a difference between Tables 3 and 11 reported in the paper? Both look almost the same. Additionally, can you clarify the metric ∆, since the captions in Tables 3 and 11 describe ∆ differently than what is described in Section 5.5? Finally, can you please explain why ∆ is better than simply presenting the difference in performance when mediators are iteratively presented to the LLM in ascending and descending orders of significance?
>
> Table 3 and Table 11 are the same but Table 11 in the appendix has the variance. We would like to apologize for the confusion regarding the delta metric. The delta metric is indeed presenting the difference in performance when mediators are iteratively presented to the LLM in ascending and descending orders of significance. The reviewers’ understanding of the metric is correct. We have improved our wording for the metric. We have made it clearer in the submission following your suggestion.
>
> > Q3:Section 4.2 is a bit unclear. Why are two variables removed, but the LLM is only required to predict one? Additionally, Figure 2b does not depict two variables being removed, so I am unsure why two are removed.
>
> We would like to apologize, the sub-figures for Figure 2 were swapped. The paper is now updated. Two variables are removed to introduce ambiguity and increase the task's complexity, simulating more realistic scenarios where the context may contain multiple plausible candidates for the missing variable. However, the LLM is required to predict only one variable to focus on its ability to identify the most relevant missing variable based on the provided causal context. This setup ensures a more controlled and interpretable evaluation of the LLM’s performance while still challenging its reasoning capabilities. We hope this helps.
>
> > W1: The paper could be clearer, especially in the methodology section, which does not adequately motivate each task or present them naturally. The tasks became clearer only upon reading the evaluation.
>
> We have attempted to make some changes and update the definitions. We hope that it has improved readability. Do let us know if a certain part of our text is still confusing to the reviewer.

---

> > ### Author Response · Authors · 2024-11-21
> > **Response to the reviewer (2/2)**
> >
> > > W2: There is a risk of data leakage in most of the datasets used. While the authors attempt to address this limitation by presenting the causal graphs to the LLMs in a novel way if the LLMs were trained on the datasets, the underlying causal patterns could be memorized, and the benchmark’s validity could be compromised. This raises the need for further analysis to determine whether LLMs are genuinely performing causal reasoning or merely memorizing causal patterns.
> >
> > A: Thank you for highlighting the concern regarding potential data leakage in our datasets. We have addressed this issue in our original submission (Lines 517–522). Table 5 in the Appendix showcases examples of LLM-generated suggestions alongside their corresponding semantic similarity scores for missing node variables in Task 3. The non-verbatim nature of these completions indicates that the LLMs are not simply retrieving memorized patterns from the training data but generating novel hypotheses based on the provided prompts. We have also added the Neuropathic pain dataset [1] which we suspect might not have been part of the pertaining, as one needs to write Python scripts to access the dataset in Appendix C.8. We observe that the results are consistent with the performance observed in other datasets. This consistency suggests that data leakage is not significantly impacting model performance. The dataset demonstrates the feasibility of the task in realistic settings and provides further evidence against data leakage concerns. We have included these results in the manuscript. We hope this alleviates the reviewer’s concern.
> >
> > [1] - Tu, Ruibo, et al. "Neuropathic pain diagnosis simulator for causal discovery algorithm evaluation." Advances in Neural Information Processing Systems 32 (2019).
> >
> > >W3: The assumption that all causal relationships are known and correct may be unrealistic in practical applications. For example, it may be improbable to know the inbound and outbound edges of a mediator variable without knowing the variable itself. This restrictive assumption limits the benchmark’s ability to fully assess LLMs’ causal reasoning.
> >
> > We acknowledge that in practical applications, having complete and accurate knowledge of all inbound and outbound edges for variables without first identifying the variables themselves can be challenging. However, this assumption was deliberately made to simplify the benchmarking framework and focus specifically on the task of hypothesizing missing variables. By assuming correct edges among observed variables, our benchmark isolates this key challenge, allowing for controlled, reproducible evaluation of LLM capabilities.
> >
> > Unlike related works that have an open-ended evaluation scheme, our paper introduces a benchmark explicitly designed to **evaluate hypothesis generation in scientific domains**. This benchmark provides structured settings and controlled metrics to rigorously assess LLM performance. While we recognize the limitations of assuming fully specified edges, we hope this approach provides a critical first step toward developing robust tools for hypothesis generation in real-world causal hypothesis generation pipelines. We have updated our limitations section to acknowledge this.
> >
> > We would like to kindly thank the reviewer for the great time and effort in reviewing our submission, The reviewer's comments has greatly improved our submission. Please do let us know if there are any remaining questions.

---

> ### Author Response · Authors · 2024-11-27
> **Any further questions/concerns?**
>
> Dear reviewer,
>
> Thank you for your insightful questions and suggestions. We have addressed your questions in detail and updated the paper. If we have adressed all of your concerns and questions, we would appreciate it, iit it could be reflected in the scores as well. We look forward to the opportunity to discuss any further questions.
>
> Best regards,
>
> Authors

---

### Official Review · Reviewer_xbDN · 2024-11-11

**Soundness:** 2
**Presentation:** 3
**Contribution:** 2
**Rating:** 5
**Confidence:** 3

**Summary:**

The paper proposes a new task of using large language models (LLMs) to propose new causal hypotheses for scientific discovery. To this end, the paper proposes a benchmark dataset based on existing causal graphs to create four tasks to prompt the LLM to identify related variables in causal hypotheses. Various language models are benchmarked on this dataset. The paper shows that LLMs can perform well in some easier tasks but not as much in more challenging open-world settings.

**Strengths:**

* originality: the paper proposed an interesting task that demonstrates how well LLMs may do in generating causal hypotheses.
* quality: the proposed tasks are defined with various metrics that help to automatically assess the performance of the LLMs. the proposed tasks also seem reasonable to me. the proposed dataset is applied to various LLMs. Finetuning experiments and chain of thoughts are used in benchmarking the models.
* clarity: The presentation of the paper is clear.
* significance: the proposed task is orthogonal to other aspects of causality such as casual discovery.

**Weaknesses:**

While the paper proposed an interesting task to generate causal hypotheses, it is not clear to me what might be the real-world utility of the proposed task beyond the evaluation with existing causal graphs. The paper will be more convincing if the author can demonstrate additional empirical benefits of the proposed task (i.e. using LLMs to generate causal hypotheses) beyond the curated causal graphs.

**Questions:**

Does few-shot learning help improve the performance of LLMs?

---

> ### Author Response · Authors · 2024-11-21
> **Response to the reviewer**
>
> We would like to thank the reviewer for their valuable feedback. We answer their question and concern below:
>
> > Q: Does few-shot learning help improve the performance of LLMs?
>
> We thank the reviewer for suggesting a few-shot example experiment. We would like to connect the answer to this question by connecting to the challenges and observations from the fine-tuning experiment discussed in papers L463-L465 and Appendix D.
>
> Few-shot learning's effectiveness depends on the example prompts used as shots. To avoid test examples becoming part of the shots, we have to use different domains as examples. Given the complexity of the Alarm graph, we decided to use them as a prior. We performed experiments with 1-shot and 5-shots for the Mixtral 8x7b model.
>
> |             | 0-shot | 1-shot | 5-shot |
> |-------------|--------|--------|--------|
> | Cancer      | 0.41   | 0.43   | 0.46   |
> | Survey      | 0.39   | 0.38   | 0.36   |
> | Asia        | 0.66   | 0.70   | 0.72   |
> | Alzheimer's | 0.31   | 0.33   | 0.34   |
> | Child       | 0.53   | 0.55   | 0.56   |
> | Insurance   | 0.46   | 0.42   | 0.45   |
>
> We observe a performance boost in some of the datasets while a drop in the others (similar to the fine-tuning exp). Alarm was a medical dataset which means that providing more examples in a different domain might hinder the model performance. Drop in performance when changing domain for in-context learning has been discussed in [1] and [2].
>
> [1] - Kwan, Wai-Chung, et al. "MT-Eval: A Multi-Turn Capabilities Evaluation Benchmark for Large Language Models." EMNLP (2024).
>
> [2] - Gupta, Akash, et al. "LLM Task Interference: An Initial Study on the Impact of Task-Switch in Conversational History." EMNLP (2024).
>
> > While the paper proposed an interesting task to generate causal hypotheses, it is not clear to me what might be the real-world utility of the proposed task beyond the evaluation with existing causal graphs. The paper will be more convincing if the author can demonstrate additional empirical benefits of the proposed task (i.e. using LLMs to generate causal hypotheses) beyond the curated causal graphs.
>
> **Significance of the work:**
>
> As mentioned in L41-44 and L58-61, our work is orthogonal and independent of causal discovery and inference. Hypothesizing causal variables in a partial graph is a predecessor to causal discovery and is heavily dependent on expert knowledge. To the best of our knowledge, this specific problem has not been previously studied. In real-world scenarios, generating hypotheses about missing variables is essential for initiating causal pipelines. For instance: in medical research, hypothesizing mediators can guide experimental design.
>
> **Regarding the curated graphs:**
>
> The graphs used in the benchmark were borrowed from realistic settings such as the Alzheimers disease graph. These graphs represent real-world causal systems across various domains, including healthcare[3], and social sciences[4]. We use curated graphs for the following reasons:
>
> - These graphs allow us to evaluate the feasibility and reliability of LLMs in generating causal hypotheses under realistic scenarios in a controlled setting. For eg: the Alz dataset captures causal relationships verified by domain experts, providing a grounded basis for testing LLMs’ ability to hypothesize plausible missing variables.
> - Including ground truth graphs allows us a point to evaluate the LLM hypothesis which is extremely difficult in an open-ended setting.
> - For a reproducible and quantifiable setup to measure, our benchmark focused on established causal graphs for the benchmark. This allows us to present a benchmark for LLM + scientific discovery evaluations.
>
> [3] Abdulaal, Ahmed, et al. "Causal Modelling Agents: Causal Graph Discovery through Synergising Metadata-and Data-driven Reasoning." The Twelfth International Conference on Learning Representations. 2023
>
> [4] VanderWeele, Tyler J., and Nancy Staudt. "Causal diagrams for empirical legal research: a methodology for identifying causation, avoiding bias and interpreting results." Law, Probability & Risk 10.4 (2011): 329-354..
>
> We would like to thank the reviewer for their questions which improved the quality of our submission. We have included the above points to highlight the significance of the paper. We would be happy to answer any potential questions from the reviewer.

---

> > ### Author Response · Authors · 2024-11-27
> > **Any further questions or concerns?**
> >
> > Dear reviewer,
> >
> > Thank you for your insightful questions and suggestions. If we have addressed all of your concerns and questions, we would appreciate it, if it could be reflected in the scores as well. We look forward to the opportunity to discuss any further questions.
> >
> > Best regards,
> >
> > Authors

---

### Author Response · Authors · 2024-11-21
**Global response**

We thank the reviewers for their valuable feedback. They appreciate the paper's originality and novelty (xbDN, oVmw, QiJv) in using LLMs to generate causal hypotheses and commend the thoughtful design of the proposed tasks and benchmark dataset. The clear and accessible presentation was highlighted as a strong point by QiJv. Reviewers also recognized the significance of this work in the fields of causality and scientific discovery, noting its potential to inspire future research on enhancing LLMs' causal reasoning abilities (1syo, QiJv). Additionally, the comprehensive evaluation across multiple LLMs and diverse domains was appreciated (oVmw, 1syo, QiJv). We are grateful for the acknowledgment of our extensive experiments and the contribution our benchmark makes to the community.

We would like to note the following modifications made to our submission:

- Added new results for a new - neuropathic pain dataset that addresses data leakage.
- Added new results for few-shot experiment (in addition to finetuning experiments part of the Appendix).
- Improved definition for MIS and the Delta metric.
- Improved definition for in-context and out-of-context distractors.
- Updated the limitations section.
- Updated some of the notations.

We further respond individually to each of the reviewers.

Best regards,

Authors

---

### Meta-Review · Area_Chair_PGKN · 2024-12-20

**Metareview:**

This paper proposes a novel benchmark for evaluating the ability of large language models (LLMs) to hypothesize missing variables in causal directed acyclic graphs (DAGs), simulating the scientific discovery process. The paper has some merits, such as its originality in proposing a new task that demonstrates the potential of LLMs in generating causal hypotheses. However, the reviewer raised some critical issues regarding data leakage and evaluation metrics.

**Additional Comments On Reviewer Discussion:**

Although the paper has some merits, such as its originality in proposing a new task that demonstrates the potential of LLMs in generating causal hypotheses (xbDN), the issues raised by the reviews are critical. For instance, the real-world utility of the proposed task is unclear (xbDN), the risk of data leakage and the restrictive assumption (oVmw), and the evaluation metrics introduced by the authors lack clarity (1syo). Although the author addresses some issues in responses, the paper still needs a major revision before it can be accepted, especially in addressing the practical utility of the task, and the clarity and validity of the evaluation metrics.

---

### Decision · Program_Chairs · 2025-01-22

Reject